# A neural basis for brain leptin action on reducing type 1 diabetic hyperglycemia

Shengjie Fan[1,2,9], Yuanzhong Xu [2,9], Yungang Lu [2], Zhiying Jiang[2], Hongli Li[2], Jessie C. Morrill[2,3], Jing Cai[2,3], Qi Wu [4], Yong Xu [4], Mingshan Xue [5,6], Benjamin R. Arenkiel [7], Cheng Huang[1✉] & Qingchun Tong [2,3,8✉]

Central leptin action rescues type 1 diabetic (T1D) hyperglycemia; however, the underlying mechanism and the identity of mediating neurons remain elusive. Here, we show that leptin receptor (LepR)-expressing neurons in arcuate (LepR$^{Arc}$) are selectively activated in T1D. Activation of LepR$^{Arc}$ neurons, Arc GABAergic (GABA$^{Arc}$) neurons, or arcuate AgRP neurons, is able to reverse the leptin's rescuing effect. Conversely, inhibition of GABA$^{Arc}$ neurons, but not AgRP neurons, produces leptin-mimicking rescuing effects. Further, AgRP neuron function is not required for T1D hyperglycemia or leptin's rescuing effects. Finally, T1D LepR$^{Arc}$ neurons show defective nutrient sensing and signs of cellular energy deprivation, which are both restored by leptin, whereas nutrient deprivation reverses the leptin action. Our results identify aberrant activation of LepR$^{Arc}$ neurons owing to energy deprivation as the neural basis for T1D hyperglycemia and that leptin action is mediated by inhibiting LepR$^{Arc}$ neurons through reversing energy deprivation.

[1] School of Pharmacy, Shanghai University of Traditional Chinese Medicine, Shanghai, China. [2] Brown Foundation of Molecular Medicine for the Prevention of Human Diseases of McGovern Medical School, University of Texas Health Science Center at Houston, Houston, TX, USA. [3] MD Anderson Cancer Center & UT Health Graduate School for Biomedical Sciences, University of Texas Health Science at Houston, Houston, TX, USA. [4] Children's Nutrition Research Center, Department of Pediatrics, Baylor College of Medicine, One Baylor Plaza, Houston, TX, USA. [5] Department of Neuroscience and Department of Molecular and Human Genetics, Baylor College of Medicine, Houston, TX, USA. [6] The Cain Foundation Laboratories, Jan and Dan Duncan Neurological Research Institute at Texas Children's Hospital, Houston, TX, USA. [7] Department of Molecular and Human Genetics and Department of Neuroscience, Baylor College of Medicine, and Jan and Dan Duncan Neurological Research Institute, Texas Children's Hospital, Houston, TX, USA. [8] Department of Neurobiology and Anatomy of McGovern Medical School, University of Texas Health Science Center at Houston, Houston, TX, USA. [9] These authors contributed equally: Shengjie Fan, Yuanzhong Xu. ✉email: chuang@shutcm.edu.cn; Qingchun.tong@uth.tmc.edu

Patients with type 1 diabetes (T1D) suffer greatly from uncontrolled hyperglycemia, and insulin delivery remains the only treatment option, but is accompanied with obesity development and life-threatening hypoglycemia. Emerging studies have demonstrated that, in addition to its role in body weight regulation, adipocyte-derived leptin is capable of reducing T1D hyperglycemia in an insulin-independent manner[1,2]. Notably, leptin action in the brain alone is sufficient to restore euglycemia in T1D, while its action in the liver is not required[3,4]. It is generally accepted that brain leptin action on reducing T1D hyperglycemia is mediated by suppressing counter-regulatory responses (CRR)[5]. The CRR includes glucagon, HPA axis, and sympathetic nerve output, representing a major physiological response against hypoglycemic for glucose homeostasis, but is aberrantly activated in T1D. For brain leptin action on reducing glucose in T1D, a role for β-adrenergic receptor-mediated sympathetic nerve output has been ruled out[6,7]. Heightened glucagon action has been suggested to be essential[8], which is later suggested to be dependent on insulin action[9]. The HPA axis has also been suggested to be the mediator[10,11], but also is later suggested not to be a sole mediator[12,13]. It is generally thought that multiple counter-regulator responses are involved and suppression of one of them may not be sufficient in mediating the leptin action[5,12].

Previous studies have shown that leptin receptor (LepR) expression in GABAergic neurons mediates leptin action in reducing hyperglycemia in T1D[6,14]. In line with this, LepR neurons in the ventromedial hypothalamus (VMH), a major brain region containing mainly glutamatergic neurons and known to regulate glucose homeostasis, are not required for the leptin action[15]. However, as both LepR and GABAergic neurons are widely distributed in the brain, the location of GABAergic LepR neurons that mediates leptin's actions on T1D glucose handling remains elusive. Specifically, conflicting results have been obtained regarding leptin action on T1D glucose in the arcuate nucleus (Arc), a region important for glucose homeostasis and known to express abundant LepRs. The melanocortin pathway, which involves proopiomelanocortin (POMC) and agouti-related protein (AgRP) neurons, two well-characterized groups of neurons in the Arc, has been shown to play an important part in mediating leptin action on T1D glucose[16,17]. Although POMC neuron function contributes only marginally, LepRs in AgRP neurons, a group of GABAergic neurons in the Arc, have been shown to contribute significantly to the leptin action[6,18]. Consistently, LepR expression in AgRP neurons greatly reduces hyperglycemia-associated leptin deficiency[19]. In contrast, chemogenetic manipulations of AgRP neuron activity revealed no effects on the leptin action on T1D glucose[20]. Importantly, as AgRP neurons only constitute a relatively small number of the Arc neuron population[21,22], the relative importance of AgRP versus non-AgRP neurons in mediating leptin action on T1D glucose remains untested.

Despite potent effects of leptin action on reducing T1D hyperglycemia, the underlying cellular mechanism remains unclear. The LepR–STAT3 signaling pathway, known to mediate the major part of leptin action on body weight and glucose homeostasis, is required for leptin action in reducing T1D hyperglycemia[7], although other signaling pathways may also play a role[23]. Leptin either activates or inhibits LepR neurons, depending on the types of neuron and their brain location[24–26]. Specifically, most LepR$^{Arc}$ neurons are inhibited by leptin[27,28], although studies focusing on POMC neurons showed that a subset of POMC neurons are activated by leptin[25,29], which may be mediated by direct action or indirectly through leptin inhibition of putative presynaptic inputs onto POMC neurons[21]. It is unknown whether leptin activation or inhibition on LepR neurons mediates its action on T1D glucose. Although leptin action on neuron activation is postulated to be mediated through TRPC

channels[30,31], little is known on the cellular mechanisms underlying leptin action on neuronal inhibition related to nutrient supply.

This study shows that LepR$^{Arc}$ neurons are aberrantly activated in T1D models, and that this activation is reversed by leptin. Specific activation of LepR$^{Arc}$, Arc AgRP neurons, or GABA$^{Arc}$ neurons reverses or abrogates leptin action on T1D glucose handling. Conversely, specific inhibition of GABA$^{Arc}$ neurons causes a leptin-mimicking effect on reducing T1D hyperglycemia, while inhibiting AgRP neurons has a much more attenuated effect. The presence of AgRP neurons is not required for T1D hyperglycemia or for the observed leptin action. Importantly, our results suggest that LepR$^{Arc}$ neurons in T1D are in a state of energy deprivation and not sensitive to nutrient deprivation, both of which are reversed by leptin. Leptin action on reducing T1D hyperglycemia is reversed by nutrient deprivation or activation of nutrient deprivation signaling pathways. These results identify aberrant activation of LepR$^{Arc}$ neurons resulting from loss in response to nutrient supply as one major central mechanism underlying T1D hyperglycemia, and reveal that leptin reduces T1D hyperglycemia through inhibition of these neurons via reversing energy deprivation.

## Results

**Aberrant activation of LepR$^{Arc}$ neurons in T1D models**. T1D exhibits a severe reduction in leptin levels[1,2], suggesting a significant contribution of loss of brain leptin action to the pathogenesis of T1D hyperglycemia. Given the known leptin effect on neuron activity, we explored potential changes in LepR neuron activity in T1D and its modulation by leptin. We examined the expression of c-Fos, a well-established indicator for neuron activity. LepR$^{Arc}$ neurons in mice with severe hyperglycemia and insulin deficiency induced by STZ, a commonly used type 1 diabetes model (STZ-T1D), exhibited greatly increased c-Fos expression in the Arc compared to controls (non-T1D conditions) (Fig. 1a, b, d), even fed ad libitum. Of note, STZ-induced c-Fos expression was largely colocalized in LepR-expressing neurons in the Arc (Fig. 1b and Supplementary Fig. 1), whereas negligible colocalization was found in other brain sites with LepR expression (Supplementary Fig. 1). The c-Fos expression in LepR$^{Arc}$ neurons was significantly reduced by i.c.v. infusion of leptin (Fig. 1c, d), suggesting a potent inhibitory action by leptin on these neurons. In addition, we also observed similar increases in c-Fos and responses to leptin treatment in another, nonobesity diabetic (NOD) model (Fig. 1e and Supplementary Fig. 2), suggesting that Arc neuron activation and its reversal by leptin represent a common feature of T1D. Interestingly, we noticed that fasting-induced c-Fos expression in the Arc was also significantly colocalized with LepR neurons (Fig. 1g, i and Supplementary Fig. 1), compared to control-fed conditions (Fig. 1f, i). Strikingly, leptin administration reduced c-Fos expression in the Arc (Fig. 1h, i), in a similar fashion to T1D. Given low leptin as a common feature of both fasting and T1D, these results suggest that LepR$^{Arc}$ neuron activation is due to reduced leptin action. Since leptin action on reducing T1D hyperglycemia is known to be associated with restraining counter-regulatory hormone levels[10,12], we tested the effect of leptin i.c.v infusion on fasting responses. While glucose levels were not different between the fed ad libitum group and the saline treatment group with 8-h fasting (Fig. 1j), glucose levels in the leptin-treated group were rapidly reduced upon food removal, and reached to 2.5 mM (Fig. 1j), at which point the experiment was ended to avoid severe glucopenic complications. Interestingly, compared to the saline-treated control group, glucagon, a typical counter-regulatory hormone, was also reduced by leptin (Fig. 1k). Thus, T1D and fasting share a common mechanism in which heightened activation of LepR$^{Arc}$

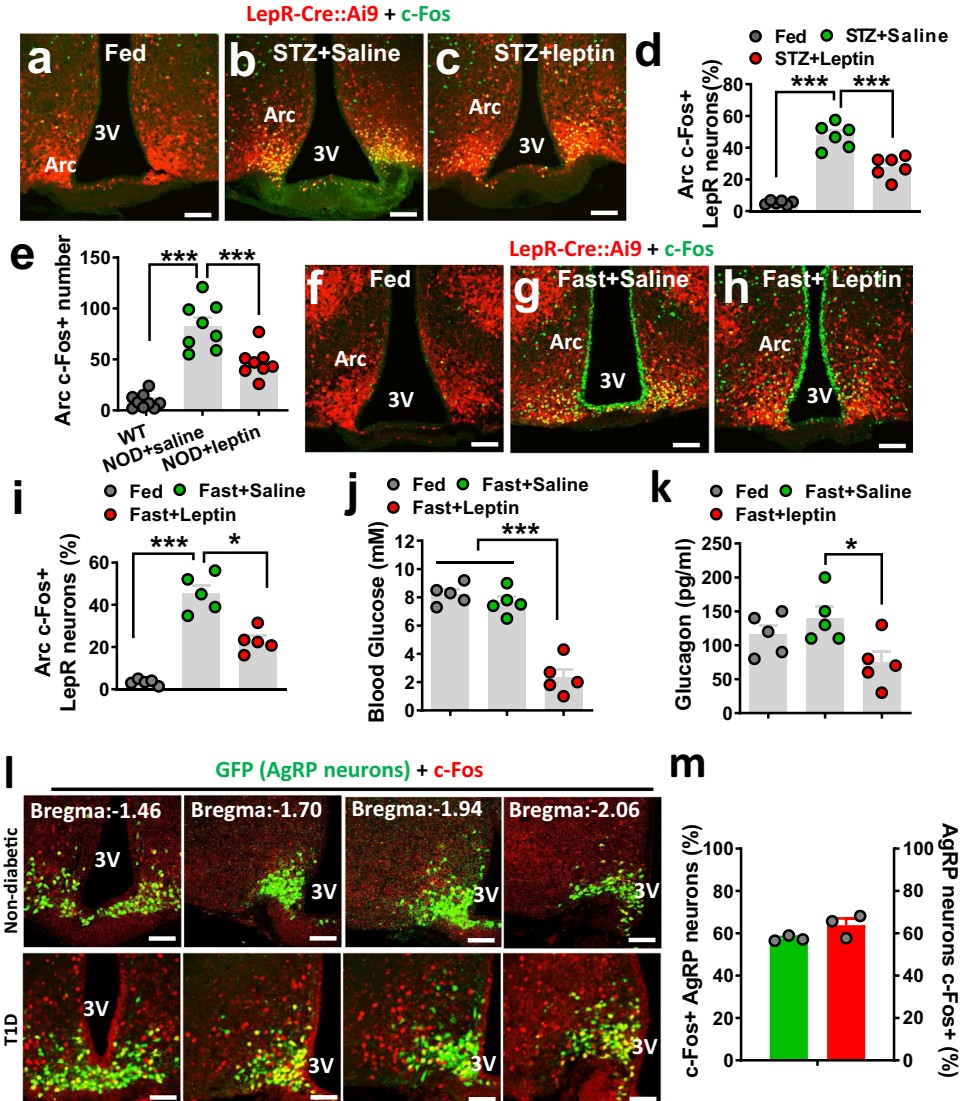

**Fig. 1 Leptin suppresses counter-regulatory responses to fasting. a–i** Leptin-Cre:Ai9 mice (8–10 weeks of age, males) were used as fed controls (**a**, **f**), made type 1 diabetic induced by STZ administration (**b**, **c**), subject to overnight fasting (**g**, **h**), or treated with i.c.v. leptin infusion (**c**, **h**). LepR neurons were shown by Ai9 reporter expression (red) and immunostaining was performed for c-Fos (green) (**a–c**, **f–h**). **d–i** Comparison in the percentage of LepR neurons with c-Fos expression among the indicated groups was shown in T1D mice (**d**, one-way ANOVA, $n = 6$/each, $F(2, 10) = 136.9$, ***$P < 0001$, control vs STZ-saline; ***$P = 0.004$, STZ-saline vs STZ-leptin) or controls (**i**, one-way ANOVA, $n = 5$/each, $F(2, 8) = 44.4$, ***$P < 0.0001$, fed vs fast-saline; *$P = 0.0035$, fast-saline vs fast-leptin). **e** Comparison in the number of c-Fos neurons in the Arc in controls, saline, and leptin-treated nonobesity diabetic (NOD) mice (one-way ANOVA, $n = 9$ for WT and $n = 8$ for both NOD-saline and NOD-leptin, $F(2, 22) = 44.69$, ***$P < 0.0001$, WT vs NOD-saline; ***$P = 0.0006$, NOD-saline vs NOD-leptin). **j**, **k** In mice presented in panels **f–i**, comparisons were shown in glucose levels between fed and fasting, and between saline and leptin treatment conditions (**j**, one-way ANOVA, $n = 5$/each, $F(2, 8) = 116.6$, ***$P < 0.0001$, fed vs fast-leptin; ***$P < 0.0001$, fast-saline vs fast-leptin) and in glucagon levels between saline and leptin-treated groups (**k**, one-way ANOVA, $n = 5$/each, $F(2, 8) = 4.575$, $P = 0.1745$, fed vs fast-leptin; *$P < 0.0427$, fast-saline vs fast-leptin). Blood glucose was measured between fed and 8-h fast (from early morning to late afternoon) and blood glucagon was measured at 8-h fasting. **l**, **m** In AgRP-Cre GFP-reporter mice, colocalization of c-Fos induction in STZ-T1D was shown in GFP-positive neurons in a series of sections from rostral to caudal Bregma levels (**l**) and percentage of c-Fos neurons that were colocalized in AgRP and percentage of AgRP neurons were c-Fos positive in T1D mice (**m**, $n = 3$/each). 3V the third ventricle, Arc arcuate nucleus, scale bar: 100 μM. All data presented as mean ± SEM.

neurons owing to reduced leptin action causes augmented counter-regulatory responses.

As AgRP neurons represent a subpopulation of Arc GABAergic (GABA$^{Arc}$) neurons[21,22] and have been a research focus on leptin action in T1D glucose, we also examined c-Fos expression patterns in these neurons in the STZ-T1D model. Using AgRP-Cre GFP-reporter mice to visualize AgRP neurons, we found that T1D-induced c-Fos expression was partly colocalized with AgRP neurons but also found in a large number of non-AgRP neurons (Fig. 1l, bottom panels), whereas a negligible number of c-Fos

neurons was observed in control non-T1D mice (Fig. 1l, top panels). Roughly 50% of c-Fos neurons were found in AgRP neurons and >60% AgRP neurons were positive for c-Fos (Fig. 1m). These results suggest potential importance for both Arc AgRP and non-AgRP neurons in mediating the leptin action.

**Leptin action on reducing T1D hyperglycemia is reversed by Arc neuron activation.** Given the abundant T1D-induced c-Fos expression in the Arc, we tested the effect of direct Arc neuron

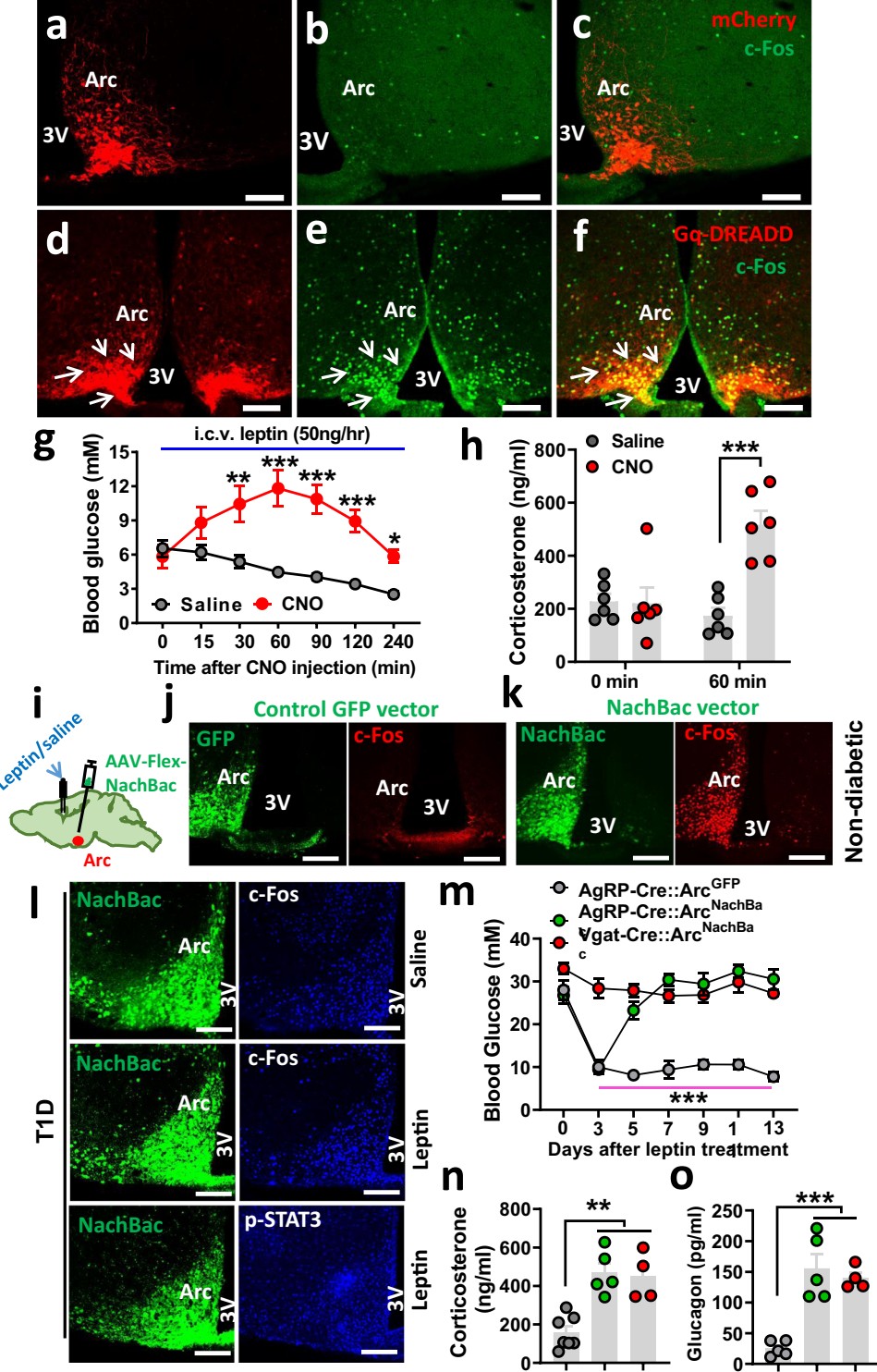

activation. We first examined whether acute activation of Arc LepR neurons reverses the leptin action. Toward this, we stereotaxically delivered control vectors (Fig. 2a) or AAV-Flex-hM3D(Gq)-mCherry vectors (Gq-DREADD) (Fig. 2d) bilaterally to the Arc of LepR-Cre mice. Selective viral targeting was verified by mCherry expression (Fig. 2a, d), and the function of Gq-DREADD was verified by CNO-induced c-Fos specifically in Arc LepR neurons (Fig. 2b, c, e, f). These mice were first made T1D and i.c.v.-treated with leptin, and then these glucose-normalized mice were divided in two groups, one treated with saline and the other with i.p. CNO. CNO effectively increased glucose compared

to falling glucose levels in controls (Fig. 2g). This glucose-increasing effect lasted for nearly 4 h, within the range of duration in CNO-induced activation of DREADD receptors. CNO-induced glucose increase was accompanied by increased corticosterone (cort) levels (Fig. 2h), an opposite response to leptin treatment effects in T1D. To rule out a potential nonspecific effect of CNO in the observed glucose-increasing effects, we generated a new group of STZ-T1D mice with glucose normalized by i.c.v. leptin treatment (Supplementary Fig. 3a), and found that CNO injections had no effect on the glucose level in these mice (Supplementary Fig. 3b). Thus, acute activation of Arc LepR neurons is

**Fig. 2 Activation of Arc neurons reverses leptin action on reducing T1D hyperglycemia. a–f** DREADD-mediated acute activation of Arc LepR neurons reverses the leptin action. LepR-Cre mice were injected with AAV-Flex-mCherry control vectors (**a–c**) and AAV-DIO-h3MD(Gq)-mCherry (**d–f**) to bilateral Arc, and expression of mCherry (**a**, **d**) and CNO-induced c-Fos (**b**, **e**) were confirmed in the Arc. Representative pictures of 6 repeats as shown in **g** and **h**. **g**, **h** Effects on glucose with single i.p. injection of CNO in T1D LepR-Cre mice with Gq-DREADD delivery to bilateral Arc and euglycemia restoration via i.c. v. leptin infusion (**g**, two-way ANOVA, $n = 6$/each, $F_{(1, 70)} = 79.47$, ***$P < 0.0001$ (at the 60-min time point); **$P = 0.0001$ (at the 120-min time point), and comparison in glucagon levels between saline and leptin-treated groups (**h**, two-way ANOVA, $n = 6$/each, $F_{(1, 20)} = 13.97$, ***$P < 0.0001$, saline vs CNO at the 90-min time point). **i–n** Effects of NachBac-mediated chronic activation of Arc neurons on the leptin action. **i** Diagram showing injections of NachBac/or control GFP vectors in AgRP or GABA$^{Arc}$ neurons through the delivery of respective vectors to the Arc of AgRP or Vgat-Cre mice, and expression of GFP (green) and c-Fos (red) with one side of the Arc receiving vector injections for control GFP vectors (**j**) and NachBac vectors (**k**). **l** In Vgat-Cre mice that were made T1D and received NachBac vector injections to bilateral Arc with i.c.v. pumps infusing saline or leptin as indicated, representative NachBac expression (GFP, left panels), and c-Fos/p-STAT3 (blue, right panels) are shown. Pictures in (**j–l**) are representative of three repeats. **m** Effects on glucose levels in response to i.c.v. infusion of leptin in the indicated groups of mice (two-way ANOVA, $n = 6$ for AgRP-GFP or AgRP-NachBac, and $n = 4$ for Vgat-NachBac, $F_{(2, 91)} = 190.6$, $P = 0.9647$, AgRP-GFP vs AgRP-NachBac at day 3; ***$P < 0.0001$, AgRP-GFP vs Vgat-Nachbac at day 3). **n**, **o** Measurements and comparison of cort (**n**, one-way ANOVA, $n = 7$ for AgRP-GFP, $n = 5$ for AgRP-NachBac and $n = 4$ for Vgat-NachBac, $F_{(2, 13)} = 16.52$, **$P = 0.006$, AgRP-GFP vs AgRP-NachBac; **$P = 0.0017$, AgRP-GFP vs Vgat-NachBac) and glucagon (**o**, one-way ANOVA, $n = 5$ for AgRP-GFP and AgRP-NachBac, and $n = 4$ for Vgat-NachBac, $F_{(2, 11)} = 21.89$, ***$P = 0.0002$, AgRP-GFP vs AgRP-NachBac; ***$P = 0.001$, AgRP-GFP vs Vgat-NachBac) levels in the blood at day 13 of i.c.v. leptin infusion shown in panel **m**. 3V the third ventricle, Arc arcuate nucleus, scale bar: 100 μM. All data presented as mean ± SEM.

sufficient to cause a transient blocking effect on leptin in reducing glucose and counter-regulatory responses.

We next examined the effect of chronic activation of Arc neurons on leptin action. Toward this, we bilaterally delivered an AAV-EF1a-Flex-EGFP-P2A-mNachBac vector to the Arc (Fig. 2i). The vector conditionally expresses, in a Cre-dependent manner, a type of sodium channel from bacteria (NachBac) that has a low membrane threshold for action potential firing and slow inactivation kinetics, and has been used to effectively elevate neuron activity without a need of ligand administration[32–34]. Given the technical difficulty in achieving specific targeting Arc LepR neurons with stereotaxic delivery to the Arc of LepR-Cre mice owing to abundant Cre expression in the nearby regions including VMH, we switched to Vgat-Cre mice, which have been used previously to specifically target Arc neurons and avoid surrounding glutamatergic neurons[32]. As leptin action on T1D glucose is known to be mediated by GABAergic neurons[6], using Vgat-Cre will be able to capture Arc neurons that mediate the leptin action. Delivery of NachBac vectors to one side of the Arc of Vgat-Cre mice led to the abundant expression of c-Fos in the injected side compared to the noninjected side (Fig. 2k), where no differences were observed in mice with control GFP vector injections (Fig. 2j), confirming that NachBac expression led to increased neuron activity levels. We then generated STZ-induced T1D in AgRP-Cre or Vgat-Cre mice. After establishing stable T1D hyperglycemia, we delivered NachBac or control vectors bilaterally to the Arc of these mice, and at the same time implanted osmotic minipumps (2-week duration) for i.c.v. infusion of leptin or saline. In contrast to severe obesity demonstrated previously[32], NachBac expression in Arc AgRP or Vgat neurons in T1D mice failed to cause body weight changes, which is likely due to already-elevated neuron activity of these neurons in the T1D condition. Consistent with the aforementioned data, in control mice with GFP vector expression, leptin treatment effectively reduced c-Fos induction in the Arc of T1D, which was associated with increased p-STAT3 expression (Supplementary Fig. 4). In contrast, leptin failed to reduce c-Fos in NachBac-expressing GABA$^{Arc}$ neurons (Fig. 2l, middle panels), compared to the saline treatment group (Fig. 2j, top panels). We validated that failure to reduce c-Fos expression was not due to failed delivery of leptin, given that we observed abundant p-STAT3 induction in the Arc (Fig. 2j, bottom rows). Similar results were also observed in NachBac-expressing AgRP neurons (Supplementary Fig. 5). Strikingly, i.c.v. leptin infusion

failed to reduce T1D hyperglycemia in NachBac-injected Vgat-Cre mice, while it maintained its normal effects in reducing glucose in GFP-injected Vgat-Cre mice (Fig. 2m). Interestingly, in NachBac-injected AgRP mice, leptin initially reduced glucose to a normal level, which was reversed around day 5 during the leptin treatment (Fig. 2m). The delayed reversal of the leptin action may reflect a delay in NachBac expression in a small number of Arc neurons (i.e., AgRP neurons) and rapid pharmacological action of leptin. Consistently, leptin infusion failed to reduce blood cort levels in the groups with failed reduction of T1D hyperglycemia (Fig. 2n). The changes in glucose were not caused by changes in body weight as there was no body weight difference among mouse groups (Supplementary Fig.6a). Consistent with the reversal of T1D-reduced glucose, the control group exhibited a significant reduction in feeding (Supplementary Fig. 6b). Together, these results demonstrate that leptin action on reducing T1D hyperglycemia is mediated through inhibition of GABA$^{Arc}$ neurons, and that aberrant activation of Arc neurons, including AgRP neurons, contributes significantly to both pathogenesis of T1D hyperglycemia and leptin action on reversal of T1D hyperglycemia.

**Acute inhibition of Arc neurons on T1D hyperglycemia.** Given the aforementioned data that when leptin failed to reduce c-Fos in the Arc it failed to reduce T1D hyperglycemia, we examined whether inhibition of Arc neurons caused a leptin-mimicking effect on reducing T1D hyperglycemia. For this, we delivered AAV-Flex-h4MGi-mCherry (Gi-DREADD) bilaterally to the Arc of AgRP-Cre or Vgat-Cre mice. To verify the effects of Gi-DREADD, we generated STZ-induced T1D in these mice. Abundant c-Fos expression was found in the Arc of control STZ-T1D mice (Fig. 3a). Consistent with differences in the number of AgRP and GABA$^{Arc}$ neurons, in CNO-injected AgRP-Cre mice, there was no obvious colocalization between c-Fos and AgRP neurons (Fig. 3b) and CNO reduced the majority of c-Fos expression in T1D Vgat-Cre mice (Fig. 3c), suggesting an effective action of CNO-induced neuron inhibition. Interestingly, a 3-day injection protocol of CNO effectively reduced T1D hyperglycemia in Vgat-Cre mice, but had no effects on either AgRP-Cre mice or "missed-targeted" mice (Fig. 3d). Notably, the reduction in glucose was transient, and the glucose levels gradually went up to control levels at day 3 after the CNO treatment protocol. No difference in body weight (Supplementary Fig. 6c) or feeding (Supplementary Fig. 6d) was observed in these mice. The

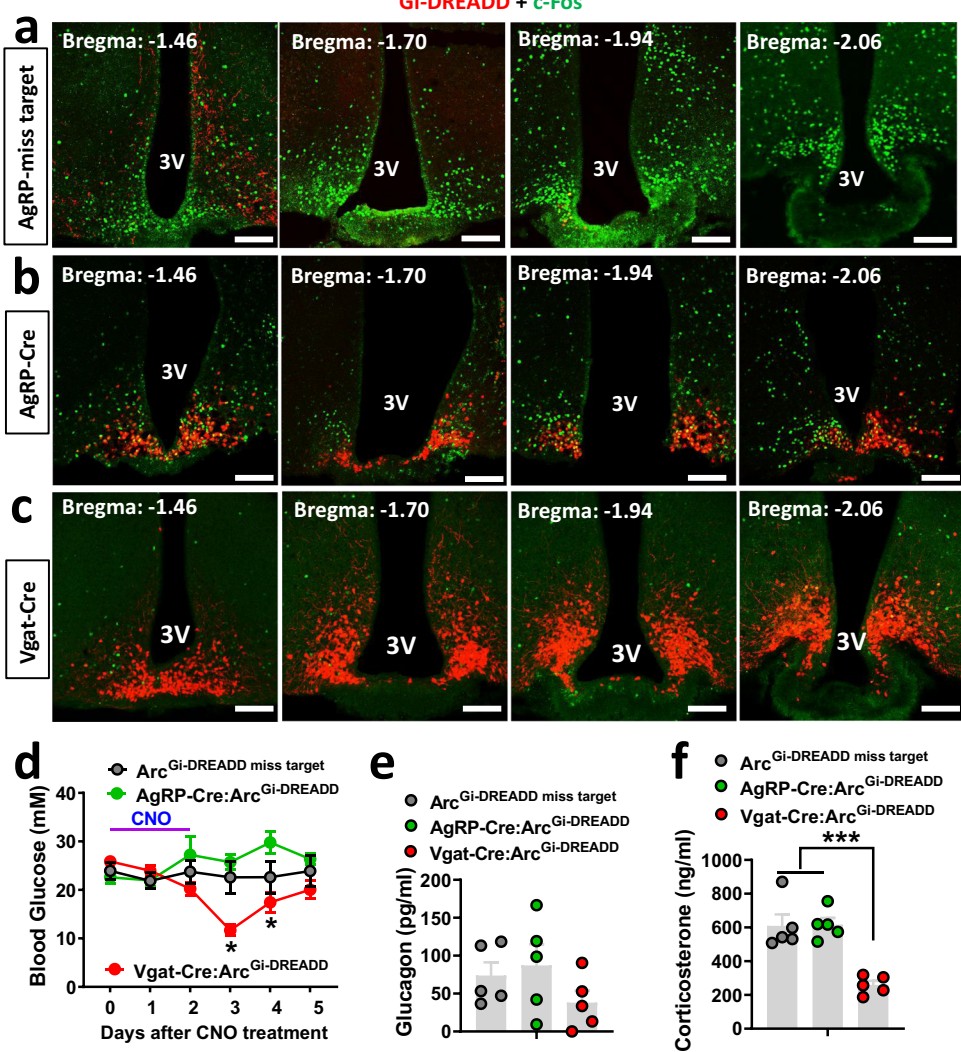

**Fig. 3 Acute inhibition of Arc neurons on T1D glucose.** AgRP-Cre or Vgat-Cre mice were injected with AAV-DIO-h4MD(Gi)-mCherry vectors to bilateral Arc and then were made STZ-T1D. **a–c** Series of sections spanning the Arc showing specific expression of viral vector expression (red) and c-Fos expression (green) in mistargeted mice (**a**), AgRP-Cre (**b**), and Vgat-Cre mice (**c**). Representative pictures over 3 repeats. **d–f** Comparisons in blood glucose in the indicated groups of mice in response to a 3-day CNO treatment protocol (twice daily for 3 days) (**d**, two-way ANOVA, $n = 5$/each, $F_{(2, 75)} = 16.41$, *$P < 0.0001$, AgRP-Gi-DREADD vs Vgat-Cre-Gi-DREADD at day 3; *$P < 0.0001$, missed vs Vgat-Cre-Gi-DREADD at day 3), comparisons in corticosterone (**e**, one-way ANOVA, $n = 5$/each, $F_{(2, 12)} = 1.449$, $P = 0.4774$, missed vs Vgat-Cre-Gi-DREADD at day 3) and glucagon (**f**, one-way ANOVA, $n = 5$/each, $F_{(2, 12)} = 18.93$, ***$P = 0.0005$, Vgat-Gi-DREADD vs missed; ***$P = 0.0005$, Vgat-Gi-DREADD vs AgRP-Gi-DREADD) at day 3 after the initiation of CNO treatment. 3V the third ventricle, Arc arcuate nucleus. All data presented as mean ± SEM. Scale bar: 100 μM.

reduction in glucose was associated with reduced corts (Fig. 3d) and a trend in reduced glucagon levels (Fig. 3e). Thus, acute inhibition of Arc GABA + neurons, but not AgRP neurons, is sufficient to cause a transient reduction in both T1D glucose and counter-regulatory responses.

**Chronic inhibition of Arc neurons on T1D hyperglycemia.** Brain leptin infusion effectively reduces T1D hyperglycemia during the whole period of leptin infusion, while acute inhibition through CNO-Gi-DREADD only produced a transient reduction in T1D glucose, suggesting chronic inhibition of Arc neurons is required to achieve long-term remission on T1D glucose. To specifically examine this, we used conditional AAV-DIO-Kir2.1-P2A-dTomato (DIO-Kir2.1) vectors expressing a mutated form of the Kir2.1 channel, which is known to effectively reduce neuron activity and excitability[32–34]. To verify Kir2.1 expression

and function, we first delivered the virus to one side of the Arc (Fig. 4a, left panels) of AgRP-Cre (top panels) or Vgat-Cre mice (bottom panels). In T1D, as expected, the noninjected side exhibited abundant c-Fos, while the injected side showed significantly reduced levels of c-Fos expression (Fig. 4a, middle and right panels). These results suggest that Kir2.1 expression leads to chronic neuron inhibition.

Using STZ-T1D AgRP-Cre or Vgat-Cre mice, we bilaterally delivered either the Kir2.1 or control mCherry viruses to the Arc, and monitored weekly glucose levels following viral injection. While control groups exhibited persistent T1D hyperglycemia, Kir2.1 expression in GABA^Arc neurons effectively reduced T1D to a normal level and this effect remained during the whole 8-week monitoring period (Fig. 4b). Interestingly, Kir2.1 expression in AgRP neurons caused an intermediate effect (Fig. 4b). Consistent with the glucose effect, feeding was reduced in both groups of mice (Fig. 4c), but body weight showed no obvious

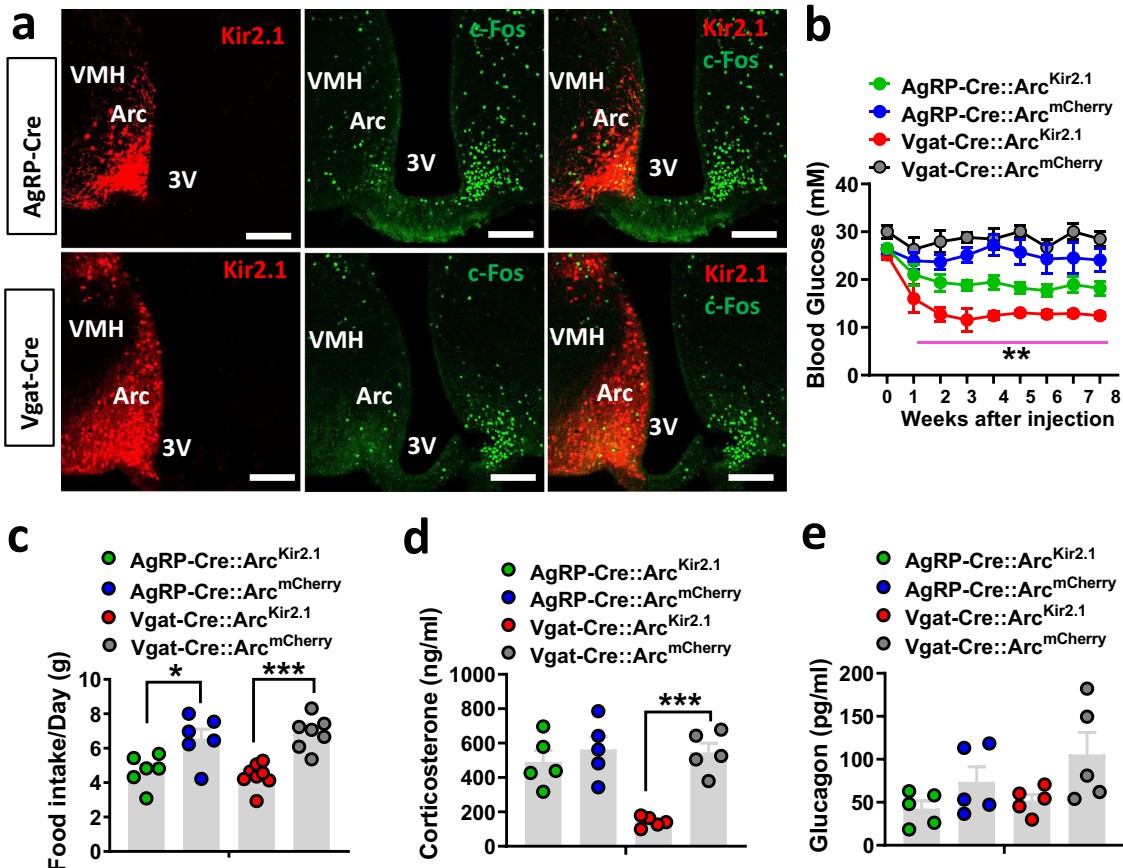

**Fig. 4 Chronic inhibition of Arc neurons on T1D glucose.** AgRP-Cre or Vgat-Cre mice (8–10 week-old males) were first made STZ-T1D and then injected with AAV-DIO-Kir2.1-P2A-dTomato vectors or control AAV-Flex-mCherry to bilateral Arc. **a** Representative sections showing comparison in the expression of Kir2.1 (red) and c-Fos (green) in the Arc of AgRP-Cre mice (top) and Vgat-Cre (bottom), in which the Kir2.1 vector was delivered to one side of the Arc. Representative pictures from $n = 2$ mice. **b–e** Weekly glucose levels following injections of the viral vectors (**b**, two-way ANOVA, $n = 6$ for AgRP-mCherry, AgRP-Kir2.1, or Vgat-mCherry, and $n = 7$ for Vgat-Kir2.1, $F(3, 189) = 120.1$, $P = 0.0641$, AgRP-mCherry vs AgRP-Kir2.1; **$P < 0.0001$, Vgat-mCherry vs Vgat-Kir2.1), food intake during week 3–4 after viral delivery (**c**, two-way ANOVA, $n = 6$ for both AgRP-mCherry and AgRP-Kir2.1, $n = 9$ for Vgat-mCherry, and $n = 8$ for Vgat-Kir2.1, $F(3, 23) = 11.68$, *$P = 0.0149$, AgRP-mCherry vs AgRP-Kir2.1; ***$P = 0.0003$, Vgat-mCherry vs Vgat-Kir2.1) and blood levels of corticosterone (**d**, two-way ANOVA, $n = 5$/each, $F(3, 16) = 12.16$, $P = 0.8091$, AgRP-mCherry vs AgRP-Kir2.1; ***$P = 0.0006$, Vgat-mCherry vs Vgat-Kir2.1) and glucagon (**e**, two-way ANOVA, $n = 5$/each, $F(3,6) = 2.904$, $P = 0.06$, Vgat-mCherry vs Vgat-Gi-DREADD at week 8 after viral injection. 3V the third ventricle, Arc arcuate nucleus. All data presented as mean ± SEM. Scale bar: 100 μM.

changes (Supplementary Fig. 6e). Cort levels were suppressed by Kir2.1 expression in Vgat-Cre mice but not in AgRP mice (Fig. 4d), while both groups exhibited a trend of reduced glucagon levels (Fig. 4e). These results demonstrate that Kir2.1 expression in GABA$^{Arc}$, but not AgRP neurons, is sufficient to restore T1D hyperglycemia, mimicking brain leptin action on T1D glucose. Together, these results reveal an important role for both Arc AgRP and non-AgRP neurons in mediating the leptin action.

**AgRP neurons are not required for T1D hyperglycemia or for leptin action on reducing T1D glucose.** To directly examine the role of Arc non-AgRP neurons in T1D hyperglycemia, we bred Vgat-Cre with the previously established strain AgRP$^{DTR}$ mice, which can be used to lesion AgRP neurons with diphtheria toxin (DTX) administration[35]. In Vgat-Cre::AgRP$^{DTR}$ mice, DTX treatment at P3 effectively killed AgRP neurons (Fig. 5a). These mice were then treated with STZ at 7–8 weeks of age, and monitored for changes in blood glucose. Interestingly, blood glucose levels were not different between DTX and control groups, both of which exhibited typical glucose levels in T1D (Fig. 5b), suggesting that the presence of AgRP neurons is not required for the development of T1D hyperglycemia. The DTX-

treated T1D mice were then divided into two groups, one treated with saline and the other with i.c.v. leptin infusion. While saline had no effects on T1D hyperglycemia, leptin effectively reduced T1D hyperglycemia to normal levels (Fig. 5c), suggesting that the presence of AgRP neurons is not required for leptin action on reducing T1D glucose. Further measurements showed that i.c.v. leptin indeed reduced c-Fos expression (Fig. 5d), glucagon, (Fig. 5e), and cort (Fig. 5f) associated with T1D development, consistent with brain leptin effects. No changes in body weight (Supplementary Fig. 6f) but reduced feeding were observed in the leptin-treated group (Supplementary Fig. 6g), consistent with the leptin-mediated restoration of euglycemia.

**Loss of LepR$^{Arc}$ neurons in nutrient sensing in STZ-T1D and reversal by leptin.** LepR$^{Arc}$ neurons become activated upon fasting when glucose is low, suggesting they are glucose-inhibited neurons. However, these neurons in T1D, an uncontrolled hyperglycemic condition, were also activated, suggesting that LepR$^{Arc}$ neurons in T1D may be defective in nutrient sensing. To test this, we examined c-Fos expression in LepR$^{Arc}$ neurons in both control and T1D mice. Compared to fed conditions (Fig. 6a), LepR$^{Arc}$ neurons responded with dramatic c-Fos

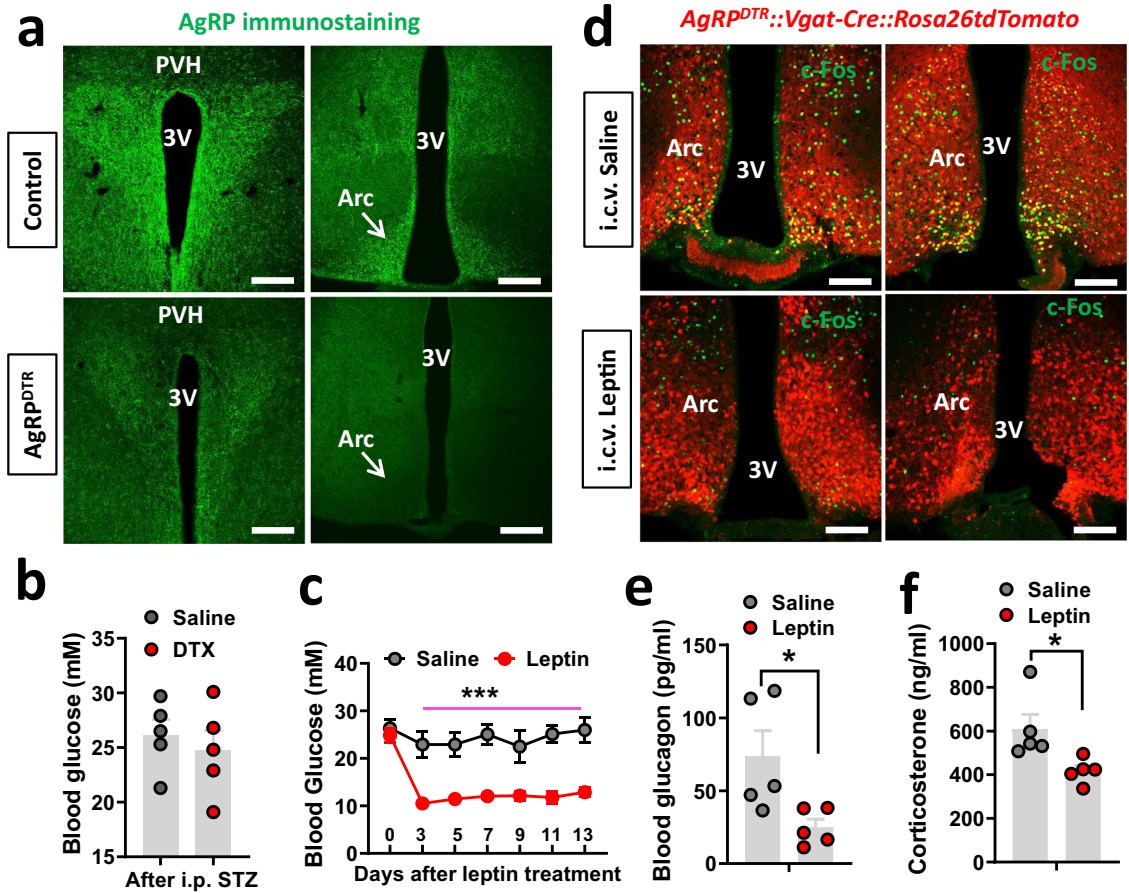

**Fig. 5 AgRP neurons are not required for T1D hyperglycemia development and for leptin action in reducing T1D glucose.** Vgat-Cre::Ai9::AgRP[DTR] mice were treated with DTX/saline at P3, made STZ-T1D at 8–9-weeks old, and then implanted with i.c.v. minipump for a 2-week leptin infusion. **a** Representative pictures from $n = 3$ mice showing immunostaining for AgRP in the Arc (right panels) and PVH (left panels) of both saline-treated controls (top) and the DTX-treated group (bottom). **b** Comparisons in blood glucose between saline and DTX-treated groups after STZ treatment (two-tailed unpaired Student's $t$ tests, $n = 5$/each, $t = 0.6019$, df = 8, $P = 0.5639$). **c** Reponses in glucose levels in DTX-treated Vgat-Cre::Ai9::AgRP[DTR] mice that received i.c.v. treatment of saline or leptin during the 14-day leptin treatment period (two-way ANOVA, $n = 5$/each, $F(1, 56) = 150.4$, ***$P < 0.0001$, saline vs leptin at day 13). **d** Representative sections from $n = 2$ mice showing expression of c-Fos (green) in Arc GABA + neurons (red) in i.c.v. saline-treated (left panel) and leptin-treated mice (right panel). **e, f** Comparisons in glucagon (**e**, two-tailed unpaired Student's $t$ tests, $n = 5$/each, *$P = 0.0266$) and corticosterone (**f**, two-tailed unpaired Student's $t$ tests, $n = 5$/each, *$P = 0.027$) at day 13 with i.c.v. leptin treatment. 3V the third ventricle, Arc arcuate nucleus. All data presented as mean ± SEM. Scale bar: 100 μM.

induction in response to fasting (Fig. 6b, e). Similar to the aforementioned results, these same neurons in T1D showed abundant c-Fos expression even fed ad libitum (Fig. 6c), and fasting failed to further increase c-Fos expression (Fig. 6d, e). As expected, blood glucose levels in the control group were significantly reduced in fasting (8.1 mM) compared to fed conditions (10.0 mM). Strikingly, glucose levels in T1D mice showed a much greater reduction (30.2–8.2 mM) in T1D mice (Fig. 6f). Thus, in T1D mice, the dramatic reduction in glucose levels (>20 mM) in response to fasting failed to initiate a significant change in c-Fos expression, which is in contrast to a dramatic increase in c-Fos expression in control mice, even with a small reduction in glucose (<2 mM). These contrasting results suggest that LepR[Arc] neurons lose the ability in nutrient sensing in vivo. To directly test this, we monitored the activity of LepR[Arc] neurons through electrophysiological recordings. For this, we primarily focused on neurons with observable changes in neuronal firing in response to altered levels of glucose and/or leptin (Supplementary Fig. 7a). With recordings from 89 Arc LepR neurons, we noted a clear effect in firing with changes in glucose and leptin (Supplementary Fig. 7b). Among them, most of them were inhibited by leptin or

glucose, and ~67% of these neurons were inhibited by both leptin and glucose (Supplementary Fig. 7b). Given our data described above on potential effects of leptin and glucose inhibition on Arc neurons in T1D glucose levels, we next chose to focus on those neurons that were inhibited by glucose or leptin.

To explore the mechanism of leptin action on nutrient sensing, we tested the response of LepR[Arc] neurons to 2-deoxy glucose (2-DG), a glucopenic agent that causes nutrient deprivation. The leptin inhibitory action on LepR[Arc] neurons was reversed by 2-DG (Fig. 6g), suggesting that the leptin inhibition requires nutrient supply. Interestingly, LepR[Arc] neurons in T1D were not sensitive to 2-DG (Fig. 6h), consistent with the notion that these neurons lose nutrient sensing. In contrast, in the presence of leptin, LepR[Arc] neurons in T1D gained inhibitory responses to increased glucose and leptin inhibition was also reversed by 2-DG (Fig. 6i), suggesting that the effect of nutrients, e.g., glucose and 2-DG, requires the presence of leptin. These results suggest that the mechanism underlying leptin action lies in facilitating nutrient sensing of LepR neurons.

To further examine the physiology associated with the reversal effect of 2-DG on leptin inhibitory action, we tested the effect of 2-DG on glucose in both control and STZ-induced T1D mice. As

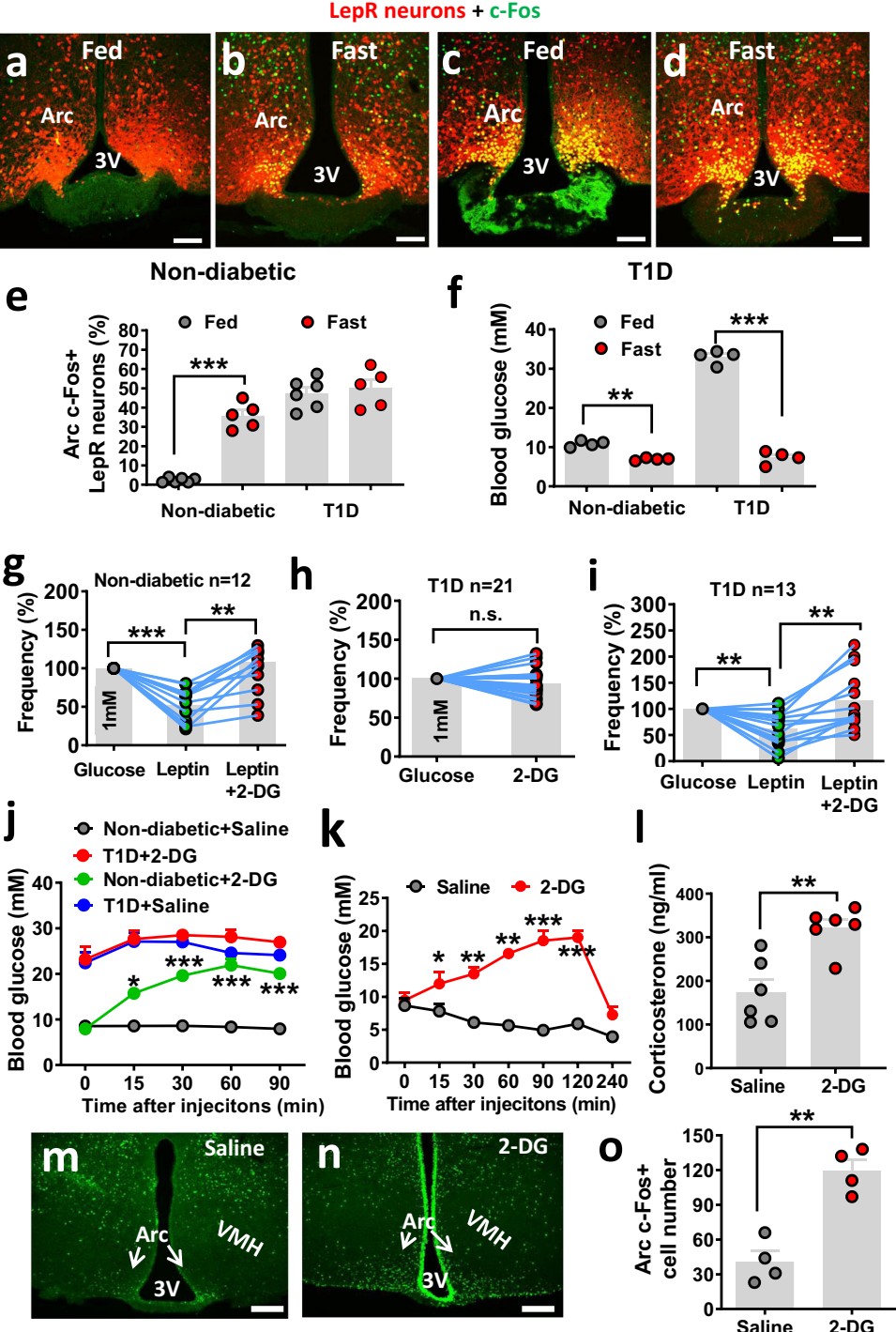

expected, i.p. 2-DG induced a significant increase in glucose levels in controls (Fig. 6j). However, it failed to further increase glucose levels in T1D mice (Fig. 6j). However, it could not be ruled out that the failure in increasing glucose levels might be due to a ceiling effect. To further address this and specifically examine the effect of 2-DG in the brain, we tested the effect of i.c.v injection of 2-DG on leptin's ability to reduce T1D hyperglycemia. We first generated STZ-induced T1D with glucose normalization by i.c.v. infusion of leptin as described previously[7]. These mice were then randomly divided into two groups, one injected with i.c.v. saline and the other with i.c.v. 2-DG. Both groups had food removed during the testing. While the saline-injected group gradually reduced glucose levels, i.c.v. 2-DG induced a rapid increase in

glucose levels (Fig. 6k), suggesting that 2-DG potently reversed the leptin action on reducing T1D hyperglycemia. The rapid increase in glucose levels was followed by a gradual drop to baseline levels at 4 h after 2-DG injection (Fig. 6k), which is consistent with a typical duration of hyperglycemia responses normally induced by 2-DG, and confirms that the rapid increase in glucose levels is not due to loss of leptin action. Interestingly, the increase in glucose levels was associated with higher corticosterone levels (Fig. 6l), suggesting that the reversal of glucose was mediated by the reversal of counter-regulatory hormone levels. Moreover, while Arc neurons in saline-treated mice showed negligible c-Fos expression (Fig. 6m, o), those in 2-DG-treated mice showed significantly more c-Fos (Fig. 6n, o).

**Fig. 6 LepR$^{Arc}$ neurons in T1D lose nutrient sensing and restoration by leptin. a–d** LepR-Cre::Ai9 reporter mice (8–10 weeks of age, males) were used either as controls (**a**, **b**) or made STZ-T1D (**c**, **d**), and were used either at fed (**a** and **c**) or overnight fasting conditions (**b**, **d**). Representative pictures from $n = 5–6$ mice showing in **e** in **a–d** showing that c-Fos immunostaining was performed and colocalized with LepR neurons. **e**, **f** Comparison in the percentage of LepR$^{Arc}$ neurons that are c-Fos positive (**e**, two-way ANOVA, $n = 6$ for non-diabetic fed and T1D fast, and $n = 5$ for non-diabetic fed and T1D fast, $F_{(1, 12)} = 36.21$, ***$P < 0.0001$, non-diabetic fed vs fast; $P = 0.7375$, T1D fed vs fast)) and blood glucose levels (**f**, two-way ANOVA, $n = 5$/each, $F_{(1, 13)} = 526.1$, **$P = 0.002$, non-diabetic fed vs fast; ***$P < 0.0001$, T1D fed vs fast) between fed and fasting or between controls and STZ groups. 3 the third ventricle, Arc: arcuate nucleus. Scale bar: 100 µM. **g–i** Electrophysiological recording was performed on LepR$^{Arc}$ neurons on brain sections from LepR-Cre reporter mice (8–10 weeks of age, males), which were used as controls (**g**, one-way ANOVA, $F_{(11, 22)} = 1.101$, $n = 13$/each, ***$P < 0.0001$, glucose vs leptin; **$P = 0.0079$, leptin vs leptin/2-DG) or made T1D (**h**, two-tailed unpaired Student's $t$ tests, $n = 21$/each, $t = 0.1622$, df = 40, $P = 0.1126$, and **i**, one-way ANOVA, $n = 14$/each, $F_{(12, 24)} = 2.285$, **$P = 0.0026$, glucose vs leptin; **$P = 0.0011$, leptin vs leptin/2-DG) and LepR$^{Arc}$ neurons were recorded for firing frequency in response to leptin and/or 2-DG. Data were expressed relative to baseline firing rates for each recorded neuron. **j** Comparison in glucose levels in nondiabetic controls and T1D mice treated with i.p. injection of either saline or 2-DG during a period of 90 min after the injection (two-way ANOVA, $n = 6$/each, $F_{(3, 50)} = 159.2$, ***$P < 0.0001$, non-diabetes—saline vs non-diabetes-2-DG, $P = 0.4716$, T1D-saline vs T1D-2-DG). **k–o** Groups of wild-type mice (8–10 weeks of age, males) were first made T1D and were then implanted with both i.c.v. minipump for leptin infusion and i.c.v. cannulas for brain injection of 2-DG or saline. Brain injections of 2-DG or saline were performed after leptin-mediated euglycemia restoration. Blood glucose was measured following i.c.v. injection of 2-DG at the indicated time points (**k**, two-way ANOVA, $n = 6$/each, $F_{(1, 35)} = 240.1$, ***$P < 0.0001$, saline vs 2-DG at the 120-min time point) and corticosterone levels were measured at 90 min after 2-DG injections (**l**, two-tailed unpaired Student's $t$ tests, $t = 4.108$, df = 10; $n = 6$/each, **$P = 0.0021$). Brain sections were used for c-Fos immunostaining from i.c.v. saline (**m**) and 2-DG treatments (**n**) at 90 min after i.c.v. 2-DG injection, and neuron number with c-Fos expression was compared between these two groups (**o**, two-tailed unpaired Student's $t$ tests, $t = 5.885$, df = 6, $n = 4$/each, **$P = 0.0011$). Data presented as mean ± SEM. 3V the third ventricle, Arc arcuate nucleus, VMH ventromedial hypothalamus. Scale bar: 100 µM.

Taken together, these results suggest that leptin inhibition of Arc LepR neurons in T1D is mediated by replenishing the nutrient supply, which is reversed by 2-DG.

**Leptin action on LepR$^{Arc}$ neurons in reducing T1D hyperglycemia involves the AMPK pathway.** AMP kinase (AMPK) is known to be a key energy sensor in neurons[36]. To test a potential involvement of AMPK in leptin-mediated glucose nutrient sensing in LepR$^{Arc}$ neurons, we performed immunohistochemistry for phospho-AMPK (p-AMPK), the activated form of AMPK that is known to be increased during energy deprivation. Compared to controls (Fig. 7a), LepR$^{Arc}$ neurons in T1D exhibited a dramatic increase of p-AMPK (Fig. 7a, b). Notably, p-AMPK expression in T1D was mostly colocalized with LepR neurons. Leptin treatment significantly reduced p-AMPK in LepR neurons (Fig. 7a, b), which was reversed by 2-DG (Fig. 7a, b) indicating a specific role for leptin action in facilitating nutrients to these neurons. Interestingly, p-AMPK and c-Fos were highly colocalized in the Arc (Fig. 7c), suggesting that activation of Arc LepR neurons is due to nutrient deprivation. To examine the role of AMPK activation in T1D hyperglycemia, we treated STZ-induced T1D mice, in which glucose levels were normalized by i.c.v. leptin infusion, with saline or AICAR, a specific agonist of AMPK. While glucose in the saline group gradually reduced during the testing period, AICAR effectively reversed the leptin action and increased glucose levels (Fig. 7d), which was associated with increased glucagon levels (Fig. 7e), consistent with a reversal of suppressing effects on counter-regulatory hormones by leptin action in reducing T1D hyperglycemia. These results suggest that leptin action on Arc LepR neuron activity involves the AMPK pathway, further confirming that leptin action on T1D glucose is mediated through suppressing the activation of LepR neurons owing to energy deprivation.

## Discussion

Insulin deficiency is the cause of T1D; however, accumulating evidence suggests an involvement of brain dysfunction in T1D pathogenesis, as demonstrated by central leptin action on euglycemia restoration independent of insulin action[3]. T1D models exhibit a severe reduction of leptin levels, suggesting a physiologic contribution of leptin loss toward uncontrolled hyperglycemia in T1D. Despite much interest in the brain mechanisms underlying the leptin action, the brain location and the nature of the defects remain largely elusive. Based on

combined mouse genetics, selective brain targeting and pharmacology, electrophysiological recordings, and the investigation of STZ-induced T1D models, our studies collectively reveal the following important insights: (1) GABAergic LepR$^{Arc}$ neurons mediate leptin action on reducing T1D hyperglycemia, (2) Arc GABAergic non-AgRP neurons play a critical role in mediating the leptin action, (3) the presence of AgRP neurons is not required for either T1D development or for leptin action on reducing T1D hyperglycemia, (4) LepR$^{Arc}$ neurons in T1D are in an activated state owing to energy deprivation, and (5) leptin action is mediated by inhibiting the activated LepR$^{Arc}$ neurons in T1D through reversing energy deprivation (Fig. 7f).

This study mainly uses a T1D rodent model induced by STZ to examine the insulin-independent effect of leptin. It is important to note that the term of T1D referred here is specifically for the state of untreated insulinopenic diabetes induced by STZ, which is distinct from the typical type 1 diabetes in humans that receive insulin treatments with largely controlled glucose levels. Our results demonstrate that GABAergic LepR$^{Arc}$ neurons mediate T1D hyperglycemia and leptin action on reducing T1D glucose. Among various LepR neurons in the brain, only LepR$^{Arc}$ neurons were selectively activated in T1D. The physiological importance of this activation is reflected from the striking effects on T1D glucose by direct manipulation of Arc neuron activity. Both acute and chronic activation of GABA$^{Arc}$ neurons reversed leptin action on reducing T1D, while inhibition of these neurons was sufficient to produce leptin-mimicking action on reducing T1D hyperglycemia. Due to technical limitations in using LepR-Cre to target LepR$^{Arc}$ neurons, which caused varied offsite targeting to nearby glutamatergic neurons in the ventromedial hypothalamus (VMH), we instead used Vgat-Cre to selectively target GABA$^{Arc}$ neurons. Since the function of LepR is mediated by GABAergic neurons in both body weight[21] and T1D glucose[6], it is most likely that GABAergic LepR$^{Arc}$ neurons mediate the effects we observed in our study. The revealed importance of GABAergic LepR$^{Arc}$ is consistent with previous observations on the noninvolvement of VMH LepR neurons in mediating leptin action on T1D glucose[15]. Thus, heightened activation of GABAergic LepR$^{Arc}$ represents one major pathogenic cause of T1D hyperglycemia. Therefore, strategies aiming to inhibit these neurons should represent an effective therapy to achieve leptin-mimicking euglycemia restoration in T1D.

Our results revealed previously unappreciated importance of non-AgRP Arc neurons in T1D pathogenesis and mediating

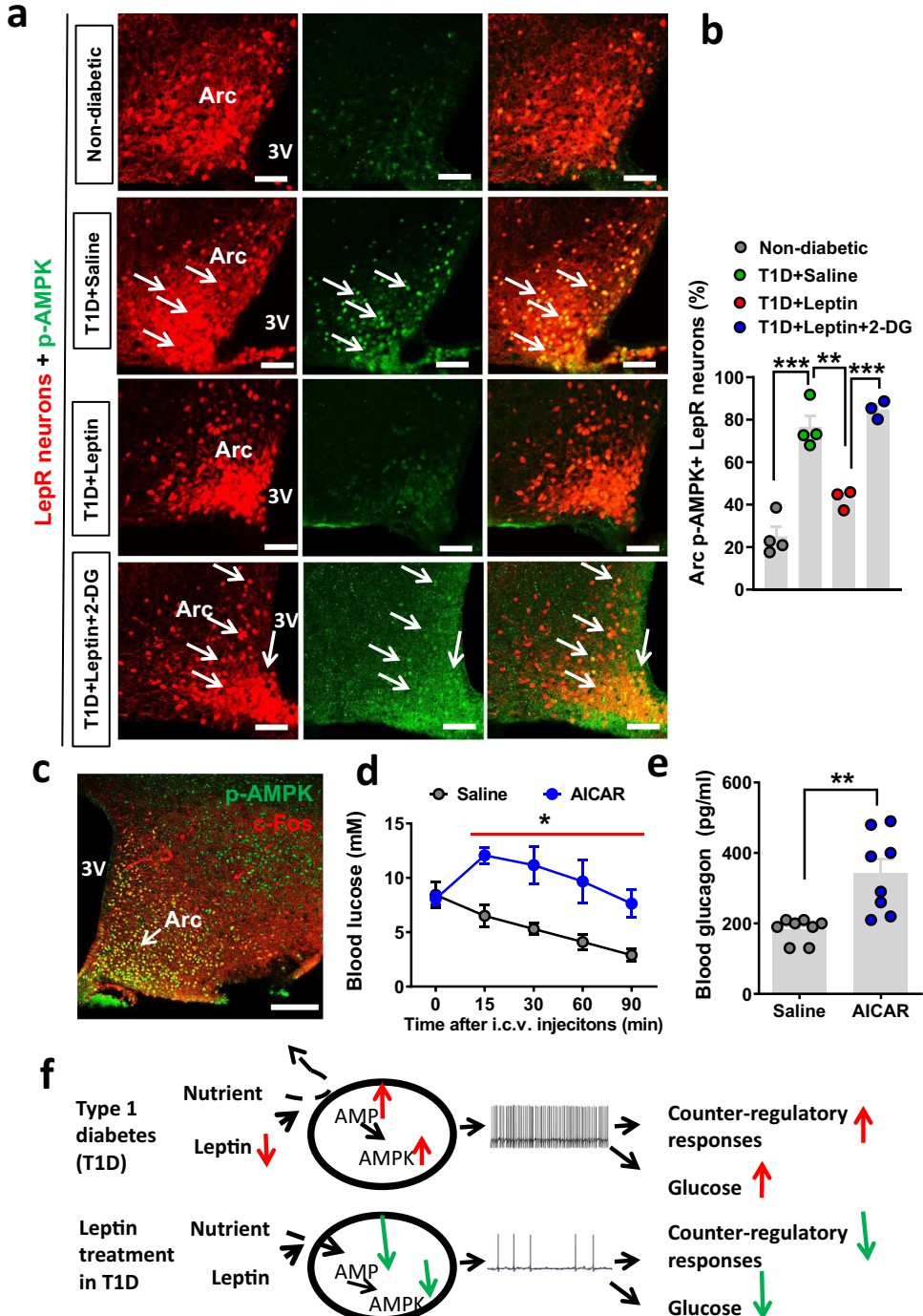

**Fig. 7 T1D hyperglycemia and leptin action on reducing T1D hyperglycemia involve the AMPK pathway. a** Representative immunostaining images from $n = 3$–4 mice as shown in **b** for p-AMPK in hypothalamic sections were shown from control (**a**, Non-diabetic), T1D with saline treatment (**a**, T1D + saline), T1D with leptin treatment (**a**, T1D + leptin), and T1D with both leptin and 2-DG treatments (T1D + leptin+2-DG) at fed conditions. **b** Comparison in the percentage of LepR$^{Arc}$ neurons with p-AMPK expression among the four groups (one-way ANOVA, $n = 4$ for nondiabetic controls and T1D-saline and $n = 3$ for T1D-leptin and T1D-leptin-2_DG, $F(3, 10) = 42.27$, ***$P < 0.0001$, nondiabetes vs T1D-saline; **$P = 0.0051$, T1D-saline vs T1D-leptin; ***$P = 0.004$, T1D-leptin vs T1D-leptin-2_DG). **c** Representative pictures from $n = 1$ mouse showing colocalization between p-AMPK (green) and c-Fos (red) in the Arc of T1D LepR-Cre mice. **d**, **e** Wild-type mice (8–9 weeks of age, males) were made T1D and then i.c.v. treated with leptin to restore euglycemia. These mice were then treated with i.c.v. injections of either saline or AICAR, an agonist of AMPK. Comparisons in blood glucose were shown following i.c.v. AICAR (**d**, two-way ANOVA, $n = 3$/each, $F(1, 20) = 35.91$, *$P = 0.0374$, saline vs AICAR at the 90-min time point) and in blood glucagon at 90 min after AICAR injection (**e**, two-tailed unpaired Student's *t* tests, $n = 8$/each, $t = 3.862$, df $= 14$; $n = 9$/each, **$P = 0.0017$). **f** Diagram depicting hyperglycemia in T1D caused by heightened Arc GABAergic neuron activity due to energy deprivation and low leptin levels (top) and euglycemia restoration by leptin inhibition of neuron activity through reversing energy deprivation (bottom). 3: the third ventricle and Arc: arcuate nucleus. Data presented as mean ± SEM. 3V the third ventricle, Arc arcuate nucleus; Scale bar: 100 μM.

leptin action on T1D glucose. Previous studies suggest the importance of the melanocortin pathway, and especially AgRP neurons, in mediating leptin action[16–19]. Given the extensive research focus on AgRP neurons, the roles for non-AgRP Arc neurons have largely been neglected. Mice with AgRP neuron lesions developed T1D hyperglycemia in response to STZ and importantly, leptin action in reducing T1D hyperglycemia in these mice remained intact, suggesting that the presence of AgRP neurons is neither required for T1D hyperglycemia nor for mediating leptin action on reducing T1D hyperglycemia. Our results showed that activation of AgRP neurons exhibited a delayed reversal of leptin action, while chronic inhibition of these neurons led to a partial reduction in T1D glucose. These findings contrast a complete reversal of leptin action and leptin-mimicking effect by respective activation and inhibition of GABA$^{Arc}$ neurons, suggesting an important role for Arc GABAergic non-AgRP neurons in mediating the leptin action. A mild role for AgRP neurons in mediating the leptin action is supported by a recent study with DREADD-mediated manipulation of AgRP neuron activity[20]. Given the demonstrated c-Fos expression in a subset of GABAergic non-AgRP Arc neurons in T1D, these Arc GABAergic non-AgRP neurons may represent an important novel brain site for T1D pathogenesis. Of note, previously identified non-AgRP Trh/Cxcl12 GABAergic neurons in the Arc may represent one such group[22]. Further delineation of the functional identity of these neurons in their underlying neural pathway may reveal new insights on leptin action on reducing T1D hyperglycemia.

We provided several lines of evidence supporting that LepR$^{Arc}$ neurons are activated by energy deprivation in T1D. LepR$^{Arc}$ neurons were similarly activated in T1D and fasting, a condition with nutritional deprivation. Importantly, brain leptin infusion was potent in reducing circulating glucose in fasting associated with suppressing counter-regulatory hormone levels, all effects mimicking leptin action on reducing T1D hyperglycemia. Supporting this, except for blood glucose levels, T1D and fasting share many common features in blood hormones and physiology, including low insulin and leptin levels, and augmented counter-regulator responses[37]. In addition, LepR$^{Arc}$ neurons in T1D show a dramatic increase in the activation of AMPK, which is known to be an intracellular energy status sensor[36]. The colocalization of p-AMPK and c-Fos in the Arc provides direct evidence supporting that energy deprivation causes LepR neuron activation. Consistently, activation of AMPK also occurs in the hypothalamus during fasting and can be reversed by leptin[36]. It is striking that the sign of energy deprivation in LepR$^{Arc}$ neurons is accompanied with uncontrolled severe hyperglycemia in T1D. Moreover, LepR$^{Arc}$ neurons failed to respond to dramatic changes in blood glucose. In contrast to abundant induction of c-Fos in control LepR$^{Arc}$ neurons in response to a reduction of <2 mM glucose in controls, no changes were observed in c-Fos in T1D LepR$^{Arc}$ neurons with a change of >20 mM glucose. These lines of evidence collectively support a state of defective nutrient sensitivity of LepR$^{Arc}$ neurons in T1D. Colocalization of c-Fos and p-AMPK in T1D LepR$^{Arc}$ neurons suggests that these neurons are "locked" in a state of energy deprivation, signaling starvation and leading to heightened counter-regulatory responses, hyperglycemia, and other changes associated with starvation, all typical symptoms of T1D.

Our data suggest that leptin action on T1D LepR$^{Arc}$ neuron inhibition and hyperglycemia is mediated by reversal of energy deprivation in T1D LepR$^{Arc}$ neurons. Leptin effectively reduces the heightened activity of LepR$^{Arc}$ neurons in T1D, in a similar fashion to that occurring in fasting states. In addition, this action is accompanied by reversing the activation of AMPK. Importantly, leptin action can be reversed by 2-DG, a nutrient-depriving agent. Conversely, T1D LepR$^{Arc}$ neurons were not sensitive to 2-DG, but

gained sensitivity in the presence of leptin, providing direct supporting evidence that leptin mediates nutrient sensitivity of these neurons. The reversal of energy deprivation by leptin is further corroborated by the effect of 2-DG in reversing leptin action on reducing T1D hyperglycemia, as well as inhibiting the heightened activity of Arc neurons, suggesting that leptin inhibition may be mediated by replenishing energy supply to these neurons. The mechanism underlying the leptin action in reversing energy deprivation is not clear, but may involve increased uptake of nutrient substrates, including blood glucose or lactate from glial cells[38,39]. Supporting this, previous observations suggest that glucose and leptin exert synergistic effects on intracellular leptin signaling[40]. In line with this, leptin action on presynaptic GABAergic input to POMC neurons is reduced by glucose and increased by the activation of AMP kinase[41], and AMPK plays a role in mediating leptin and glucose effects on LepR neurons[24,36,40,42]. Increased intracellular nutrient supply may inhibit neuron activity through membrane channels[28].

In our recording experiments, we focused on LepR$^{Arc}$ neurons that were inhibited by leptin, as these neurons are presumed to be the most relevant to T1D hyperglycemia due to their heightened activity associated with low leptin levels. The remaining leptin-excited LepR$^{Arc}$ neurons, including POMC neurons, may also be involved in glucose homeostasis[26]. However, as Arc GABAergic LepR$^{Arc}$ neurons are the primary mediator of the leptin action in T1D, POMC neurons, which are non-GABAergic[21], may contribute little to T1D hyperglycemia and leptin action in reducing T1D glucose.

Notably, a recent human clinical trial with a combination of insulin and leptin treatment of T1D shows that leptin reduces the dose of insulin for human T1D treatment, but without a dramatic effect on glucose[43]. Although this failed clinical trial, which is in contrast to the demonstrated leptin effect in rodents, may be due to many reasons, including potential differences between humans and rodents, one key difference in human T1D patients is that they are always on insulin treatment, and the leptin level is only reduced 50%, compared to non-T1D humans[5,44,45], presumably due to the reason that insulin is capable of increasing leptin levels[46,47]. In contrast, rodent T1D models are essentially insulinopenic and as a result, leptin levels are extremely low[1,2,46,48]. It is thus conceivable that an additional leptin dose on top of the mildly reduced leptin level in human T1D may not be able to provide additional beneficial effects. In other words, the beneficial effect of brain leptin action may have been reflected by the effects of insulin treatments[49]. Further studies are warranted to specifically address this possibility. In summary, our current study reveals that uncontrolled insulinopenic T1D represents a state of chronic fasting in terms of chronic activation of LepR$^{Arc}$ neurons owing to energy deprivation. Together, these data demonstrate that leptin action on reducing T1D hyperglycemia is mediated through reversing energy deprivation in LepR$^{Arc}$ neurons, leading to inhibition of these neurons, and reduced counter-regulatory responses. Thus, understanding the mechanism underlying leptin action on reversal in energy deprivation is imperative for a full understanding of T1D pathogenesis and more efficient treatment strategies against T1D.

## Methods

**Animals.** All mice were housed with ad libitum access to water and food in a temperature-controlled room (21–22 °C) with a 12:12-h light–dark cycle. Animal testing and research follow all relevant ethical regulations. Animal care and procedures were approved by the Animal Welfare Committee of The University of Texas Health Science Center at Houston (Protocol # 18-021). LepR-Cre mice were bred to (ROSA)26Sortm9(CAG-tdTomato) (also called Ai9) reporter to generate LepR-Cre::Ai9 mice for electrophysiological recording or colocalization with c-Fos immunohistochemistry. AgRP-Cre and Vgat-Cre mice were obtained from the Jax Laboratory, and these mice were also bred with (ROSA)26Sortm1.1(CAG-cas9*,-EGFP) to allow visualization of AgRP and GABAergic neurons with GFP expression (GFP reporters). All mice that were used for stereotaxic injections were at least 8–10-weeks old. AgRP$^{DTR}$ mice were provided by Dr. Qi Wu of Baylor

College of Medicine and AgRP lesion with the injection of DTX (Sigma, St. Louis, MO). The experiments on the nonobese diabetes model (NOD, Jax lab) were performed in Shanghai University of Traditional Chinese Medicine (SUTCM) and the procedures were approved by the Animal Welfare Committee of SUTCM (Approval number: SZY20160900).

**STZ-induced T1D**. STZ (Sigma-Aldrich, St. Louis, MO) dissolved in cold-sterile acetate buffer, pH 4.5, was injected intraperitoneally (i.p.) to 6–8-week-old male mice at 150 mg/Kg (two times at 1-week intervals). Nondiabetic control mice received i.p. injection of acetate buffer alone. T1D was defined as nonfasting blood glucose >23 mM on 3 consecutive days.

**Icv pump infusion studies**. Recombinant mouse leptin (A.F. Parlow, National Hormone and Peptide Program, Torrance, CA) dissolved in phosphate-buffered saline (PBS, pH 8.0) was loaded into 14-day mini-osmotic pumps (DURECT Corporation, Cupertino, CA) that were connected with commercial brain infusion kits (DURECT Corporation). Pumps were surgically implanted in the mouse interscapular cavity 7 days after the establishment of T1D. All mice were anesthetized with ketamine (100 mg/Kg) and xylazine (10 mg/Kg), and then cannulas were stereotaxically positioned into the lateral cerebral ventricles using the coordinates: - 0.30 mm from the bregma, ± 1.0 mm lateral, and −2.5 mm of the skull. Leptin was continuously delivered at the speed of 50 ng per hour (0.25 µL/h) to the ventricles. T1D mice receiving the same amount of icv vehicle infusion were taken as controls. CNO (Sigma, St. Louis, MO) was i.p. injected at the dose of 1 mg/kg.

From day 3 after icv surgery, blood glucose levels, food intake, and body weight of all animals were recorded once every two days. All measurements were performed between 10:00 AM and 12:00 AM. Glucose levels were measured by using tail vein blood with OneTouch Glucometer.

For dual intracerebroventricular cannulation surgery, T1D mice were anesthetized and then cannulas were positioned into the left lateral cerebral ventricles with the coordinates: bregma, −0.5 mm, midline, −1.8 mm, and dorsal surface, −2.2 mm at a −10° angle. Meanwhile, minipumps with leptin (0.1 mg/ml, 100 µl/pump) were implanted into the right lateral cerebral ventricles with the coordinates: bregma, −0.5 mm, midline, +1.8 mm, and dorsal surface, −2.2 mm with a + 10° angle. 2-DG (2 mg/3 µl/mouse) was i.c.v. injected after 3-day recovery while leptin was pumping into the brain as well and blood glucose levels were measured.

**Stereotaxic injections and viral vectors**. Stereotaxic surgeries to deliver viral constructs were performed based on a revised previous protocol[7]. Briefly, mice were anesthetized with a ketamine/xylazine cocktail (100 and 10 mg/kg, respectively), and their heads affixed to a stereotaxic apparatus. Viral vectors were delivered through a 0.5-µL syringe (Neuros Model 7000.5 KH, point style 3, Hamilton, Reno, NV, USA) mounted on a motorized stereotaxic injector (Quintessential Stereotaxic Injector, Stoelting, Wood Dale, IL, USA) at a rate of 30 nL/min. Viral preparations were made through the Baylor NeuroConnectivity Core and Gene Vector Core with either serotype 5, 9, or DJ and titered at more than ~$10^{12}$ particles/mL. Viral delivery was targeted to the Arc through four local injections with two each side (200 nL/side, anteroposterior (AP): −1.35 and −1.5 mm, mediolateral (ML): ± 0.2 mm, and dorsoventral (DV): −5.9 mm). AAV-DIO-hM3D(Gq)-mCherry, AAV-DIO-hM4D(Gi)-mCherry, AAV-EF1a-Flex-EGFP-P2A-mNachBac, or AAV-EF1a-DIO-Kir2.1-P2A-dTomato were delivered bilaterally into the Arc of AgRP-Cre or Vgat-Cre mice. For LepR-Cre, a volume of was injected into bilateral Arc instead. Two mutations (E224G and Y242F) were made in the Kir2.1 construct so that its activity will be less rectified by membrane potentials, thereby more effective in reducing neuron activity[50]. AAV-Flex-GFP or AAV-Flex-mCherry injections were used as a control group.

**Blood collection and tissue processing**. Blood samples were collected from all mice with deep anesthesia into a chilled tube containing 15 µl EDTA. Plasma was isolated by centrifugation (3000×g at 4 °C for 15 min), and then stored at −80 °C. After blood collection, all mice were transcardially perfused with 0.9% normal saline and followed by 10% formalin. Brain and pancreas were harvested and cryoprotected in 30% sucrose for further analysis. Plasma glucagon, insulin, and corticosterone were measured by the Vanderbilt University Hormone Assay & Analytical Services Core (Nashville, TN).

**Immunohistochemistry (IHC)**. Brain sections were sliced with microtome (Leica, Germany) at 30-µm thickness. Phosphor-STAT3 (p-STAT3) was immunostained with rabbit anti-p-STAT3 antibody (1:400, Cell Signaling, Danvers, MA), p-AMPK (Cell Signaling, Danvers, MA) used at 1:1000, and c-Fos antibody (Millipore) used at 1:1000, the specificity of which was confirmed previously[51], or all brain sections were incubated with antibodies using free-floating methods with 24-well plates[52], and visualized with AlexaFluor 488- conjugated donkey anti-rabbit IgGs (1:500, Jackson ImmunoResearch Laboratories, West Grove, PA). The sections were photographed under a TCS SP5 confocal microscope (Leica). Neurons with clear immunostaining of c-Fos or p-AMPK were counted from four to five matched sections containing Arc from individual mice (n = 3–4/group) and the number of was averaged and compared using the indicated statistical methods.

**Brain slice electrophysiological recordings**. Coronal brain slices (250–300 µm) containing the arcuate nucleus of the hypothalamus (ARC) were cut in ice-cold artificial cerebrospinal fluid (aCSF) containing the following (in mM): 125 NaCl, 2.5 KCl, 1 MgCl$_2$, 2 CaCl$_2$, 1.25 NaH$_2$PO$_4$, 25 NaHCO$_3$, and 11 D-glucose bubbling with 95% O$_2$/5% CO$_2$. Slices containing the ARC were maintained for recovery for at least 1 h at 32–34 °C. Individual slices were transferred to a recording chamber mounted on an upright microscope (Olympus BX51WI) and continuously superfused (2 ml/min) with ACSF warmed to 32–34 °C by passing it through a feedback-controlled in-line heater (TC-324B, Warner Instruments). Cells were visualized through a ×40 water-immersion objective with differential interference contrast (DIC) optics and infrared illumination. Whole-cell current-clamp recordings were made from neurons with the tdTomato reporter fluorescence within the Arc. Patch pipettes (3–5 MΩ) were filled with a K$^+$-based low Cl$^-$ an internal solution containing (in mM) 145 κ-gluconate, 1 MgCl$_2$, 10 HEPES, 0.2 EGTA, 4 ATP, 0.3 GTP, and 10 phosphocreatine, pH 7.35, 270–285 mOsm. To test the glucose dose effect on neuronal firing, the osmolarity of the solution was balanced with sucrose. Mouse leptin (Harbor-UCLA Medical Center) at a concentration of 80 nM was perfused at 2 ml/min for up to 3 min to examine the effects of leptin on LepR neurons. To test LepR$^{Arc}$ neurons from T1D mice, brain sections were obtained 1 week after the establishment of T1D conditions with blood-glucose concentration >25 mM. For recording, we used bath 1 mM as a starting glucose concentration to mimicking a low glucose level, aiming for a higher level of baseline activity for leptin inhibitory action. We increased glucose levels from 1 to 10 mM, which grossly mimics high brain glucose levels in T1D, to examine the effect of glucose on neuron activity change.

**Hormone measurement**. Cheek blood from mice was collected and the serum was stored in a −80 °C freezer until measurement. Hormone measurements were performed either by ELISA kits from Millipore or by RIA kits at the Hormone Assay Core Facility of Vanderbilt University.

**Statistical analysis**. All data were presented as mean ± SEM. Statistical analyses were performed using Student's t tests or ANOVA tests with Tukey post hoc test as specified for each study (GraphPad Prism, GraphPad Software, La Jolla, CA). A P value <0.05 was considered significant.

**Reporting summary**. Further information on research design is available in the Nature Research Reporting Summary linked to this article.

## Data availability
All data generated or analyzed during this study are included in this published article. Source data are provided with this paper.

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

## Acknowledgements

This study was supported by NIH R01 DK114279 and R21NS108091 (Q.T.), R01DK109934 and DOD W81XWH-19-1-0429 (Q.T. and B.R.A.), NIH R01 DK120858 (Q.T. and Yong X.), NIH R01DK117281 and R01DK101379 (Yong X.), NIH R01DK 109194 and USDA/CRIS grant 3095-5-001-059 (Q.W.), and NIH R01MH117089 and McKnight Foundation (M.X.). We also acknowledge the Neuroconnectivity Core funded by NIH IDDRC grant 1 U54 HD083092 and Baylor College of Medicine Gene Vector Core for providing AAV vectors. S.F. was supported by the Graduate Student Overseas Study Program (B11-1-8) from Shanghai University of Chinese Traditional Medicine. We would like to acknowledge the Tong lab members for helpful discussion, Dr. Zhengmei Mao for help with microscopy. Q.T. is the holder of Cullen Chair in Molecular Medicine at McGovern Medical School.

## Author contributions

S.F. conducted the research with help from Yuanzhong X., Y.L., Z.J., J.C.M., J.C., and H.L.; Q.W., M.X., B.R.A., and Yong X. provided essential reagents; Q.T. and C.H. conceived and designed the experiments, and wrote the paper with significant inputs from Q.W. and Yong X.

## Competing interests

The authors declare no competing interests.
