## [Peer Review File · Nature Communications]

Reviewers' Comments:

Reviewer #1:

Remarks to the Author:

Review of NCOMMS-20-07034-T A Neural Basis for Brain Leptin Action on Reducing Type 1 Diabetic Hyperglycemia

Using off-the shelf genetic models and immunohistochemistry and DREADDS this paper probes the nature of the neurons that coordinate the anti-hyperglycemic effects of leptin in STZ induced type 1 diabetes in mice.

The work shows that ARH GABAergic neurons mediate the effect of leptin, and that the effect does not require AgRP neurons. The work also suggests that nutrient sensing is defective in ARH GABAergic neurons, although I was not strongly convinced of this data, and felt it would be worth considering the effect of insulin in this part of the work.

The studies appear well conducted, and the data are well explained. The results are believable.

The discussion is appropriate with the caveat that the clinical data needs to be discussed, and they need to reiterate in the discussion that this therapy was tried in humans and did not produce clinically meaningful changes in glucose levels.

Edit to methods section: I did not see any fiber optic work, it may have been an earlier version. "Stereotaxic surgeries to deliver viral constructs and for optical fiber implantation were performed as previously described 3."

Ref 3 is: Fujikawa, T., Chuang, J. C., Sakata, I., Ramadori, G. & Coppari, R. Leptin therapy improves insulin-deficient type 1 diabetes by CNS-dependent mechanisms in mice. Proceedings of the National Academy of Sciences of the United States of America 107, 17391-17396, doi:10.1073/pnas.1008025107 (2010).

I assume this is incorrect reference?

Reviewer #2:

Remarks to the Author:

I read this with interest and there is much of merit in this manuscript.

In particular, there are a lot of elegant studies which appeared to have been performed with great technical capability and the number of different experimental approaches (some of them beautiful) have been employed.

The underlying premise is that a model of T1D (mostly streptozotocin-induced diabetes with one study also done in NOD mice) is characterised by low leptin levels which leads to activation of key neuronal populations within the arcuate nucleus.

Comments

1) Although a "presentational", I am concerned by the repeated use of T1D throughout the manuscript. This isn't T1D but a model of untreated insulopenic diabetes. Even in the NOD studies where immune mediated damage, mice are not treated with insulin replacement. I think they need to be clearer about this. This may seem pedantic but this is very different from T1D and of course this is the reason for low leptin. Parenthetically, were circulating leptin levels measured?

2) David Coppari showed some years ago that leptin acts via hypothalamic GABAergic neurones to lower blood glucose (Cell Metabolism 2013 doi: 10.1016/j.cmet.2013.08.004) so the novelty here appears to be narrowing this down to arcuate and nicely showing with DREADD that leptin inhibits target cells. Their findings are also consistent with Lowell's arcuate "census" n11.Trh/Cxcl12 neurons which were (non-AGRP) GABAergic leptin responsive population. I think both of these papers are worth incorporating

3) Scientifically, the approach is a little confusing as they have flipped from using LepR-cre to GLUT-cre. This makes this scientifically heterogenous as some of the studies are looking at arcuate leptin receptor containing neurones

- 4) Corticosterone and/or glucagon data are shown but no clear whether both were measured in all studies or whether only the hormone shown was measured and why selected?
- 5) Although I accept that some of their other data suggest that AGRP neurons not involved in this action of leptin, I am unconvinced by the DTR ablation studies (Fig 5 etc). We know from other studies that there is compensation with early life ablation of AGRP neurons resulting in no real phenotype (whereas adult ablation is rapidly fatal). My bias is to remove this study from manuscript
- 6) "The leptin inhibitory action on LepRArc neurons were reversed by 2-DG (Fig. 6g), suggesting that the leptin inhibition requires nutrient supply" If anything, figures 6 g,h and I show that leptin exerts its inhibitory effect regardless of glucose manipulations?
- 7) Should the label in fig 2 panel c in red read mCherry not Gq DREADD?
- 8) I am unconvinced by the 2DG studies. To my eye, figure 6j just shows a ceiling effect ie glucose levels and presumably stress/ counterregulatory hormones are already very high and little scope for further elevation with glucopenic challenge?
- 9) "LepRArc neurons in T1D were not sensitive to 2-DG (Fig. 6h), consistent with the notion that these neurons lose nutrient sensing". These are neurons that have been bathed in vivo in high levels of glucose but then studied ex vivo at 1 mM. My feeling is that they will already be operating under an environment they perceive as glucopenic so not surprising that no further effect of 2DG. It is difficult to know whether this is plausible as data are just presented as bar graphs normalized to 1 mM glucose conditions (designated "100%").
- 10) Studies shown in figures 6a b e are surely largely a scientific iteration of those shown in fig 1?

Reviewer #3:

Remarks to the Author:

This manuscript describes the results for experiments designed to investigate the role of leptin receptor positive arcuate neurons in T1D-mediated hyperglycemia. The experiments are well designed, the Methods utilised appropriate, figures are of excellent quality and the overall data presentation and interpretation is very good. The findings and conclusions presented herein are very interesting, novel and should have a significant impact for those interested in understanding the aetiology of diabetes and the complications associated with chronic and severe hyperglycemia. There are some minor errors and typographical mistakes to be corrected. In addition, there are some comments below that need addressing to clarify certain issues.

Main Comments:

General comments:

1. In the Introduction, define what the CRR is for readers and what constitutes the response. The CRR (which often becomes defective in many individuals with T1D) is normally associated with hypoglycemia excursions than chronic hyperglycemia and this could be made clearer in the text in relation to T1D and glycemic variability.
2. The hormones associated with the CRR are not all represented in this manuscript. Although data for glucagon and corticosterone are presented in many figures, no mention is made of epinephrine, a key component of the CRR. Did the authors monitor the plasma levels of epinephrine? If so can they present these data and if not, can they state why they did not?
3. The main mouse model of T1D that is used here is the STZ-treated animal. However, this leaves the animal with little insulin. Do the authors consider low residual insulin as having enhanced action in the presence of leptin – other studies suggest not – but could this issue be alluded to in Discussion? Also, in humans T1D would be treated by insulin therapy to reduce BG. What is the evidence that leptin would be a useful adjunct therapy under these conditions? Please add some discussion of this point.
4. Is the chronic activation of these arcuate GABA neurons simply down to the reduced plasma leptin, or will other factors/hormones associated with T1D state also contribute – such as relative lack of insulin – as insulin known to act on arcuate neurons to alter electrical activity and modify glucose homeostasis (e.g. acting on AGRP neurons).

5. Electrophysiological characterization showing primary data is presented in a Supplemental Figure where the authors show an exemplar whole-cell recording from one LepR neuron and relative numbers for excitability changes. The concentrations of glucose used in these experiments need to be explained in light of the plasma glucose levels presented in the other figures – ranging from ~6 to 30 mM. Why were 1 mM and 10 mM glucose used for these experiments. Also, readers should be reminded that brain glucose levels are much lower than plasma glucose and 1 mM would be equivalent to hypoglycemia whereas 10 mM is much higher than central neurons are likely to experience (~4.5 mM is usually the upper limit in severe hyperglycemia according to published figures).

6. In Figure 6 a-f, comparing c-fos expression in the fasted-fed transition in non-diabetic vs T1D mice. Why is BG reduced to such a large extent on fasting with a high proportion of Arc c-Fos+ neurons – which as you have shown clearly correlate with high plasma glucose? Presumably these T1D neurons are not able to undergo hyperpolarisation/reduced firing under fasting conditions (as you have argued – loss of ability to nutrient sense) – so what is causing BG to decline here? In your model you show 2DG + leptin increases firing of ARC LepR+ neurons suggesting nutrient sensing disturbance and reduced glucose metabolism giving rise to excitation. So in fasted state with diminished glucose metabolism (no leptin added?) these neurons will be excited as you show – so expect a higher BG associated with increased levels of glucagon and cort? Please explain?

7. Figure 7. Activation of AMPK associated with increased BG and in T1D, LepR+ neurons exhibit high p-AMPK. Leptin decreases p-AMPK in the T1D model – so my question is does addition of 2DG act to increase p-AMPK in Arc LepR+ neurons in leptin-treated T1D mice in association with raised BG?

8. No mention of how the effects of central leptin in the T1D model are transmitted to peripheral tissues to improve BG levels. Discuss role of central leptin pathway to ANS to alter glucose uptake in muscle and BAT as well as diminished hepatic glucose output? How might this be tested – use of a ganglion-blocker?

Minor Comments:

1. Figure 1 legend: ANOVA?
2. Page 6 L11-12: "...expression was apparent in both AgRP and non-AgRP neurons...? Please clarify what is meant here.
3. Page 8 L15-17 & Fig3a,b: (a) -AgRP (b) – Vgat The text has (a) as Vgat and (b) as AgRP
4. P9 L18-19 Fig 4d,e: In text cort levels (4d) and glucagon (4e) but in legend and figure glucagon is (d) and Cort is (e)
5. Page 10 L5-6 Fig 5b: Blood glucose levels for saline vs DTX treated T1D animals in text and figure legend but shown as saline vs leptin in figure. Please clarify.
6. Supplemental Fig 5: Representative WCR – please give an indication of Vm for these traces and comment on any changes in this parameter with altered glucose and/or by leptin?
7. Fig 7a middle image: The arrows are not aligned across the panels and there is no description relating to the meaning of the arrows in legend or text
8. P16 L1-2 : This sentence needs addressing – ... in face with ...: Unclear what is meant here. Also next sentence – LepRArc neurons lose responses...? Line 5 further

Reviewer #4:

Remarks to the Author:

The study by Fan S et al also indicates tried to reveal a neural basis for brain leptin control of type I diabetic high blood glucose levels. They reported that leptin receptor expressing GABAergic neurons in the hypothalamic arcuate nucleus were activated in T1D, similar to that by fast, and activation of these neurons were able to reverse the leptin's effects. The manuscript was well organized and the results were interpreted clearly, while the conclusions were premature, and more new studies are required to increase the impact in this study fields. Meanwhile, I have other

comments as shown in the below.

The neuron populations for leptin control of glucose in normal and diabetic animals have already revealed by previous studies (i.e. Fujikawa T et al., *Cell Metabolism*, 2013) including GABAergic and POMC neurons in the hypothalamic arcuate nucleus. The authors reported that leptin receptor expressing neurons in T1D animals were activated, which might be attributable to decreased leptin. In this study, plasma leptin should be detected in the experimental conditions. Meanwhile, it is also important to examine the mRNA and/or protein expressions of leptin receptors in the identified neuron populations.

Figure 1:

In Figure 1a-c, in addition to the arcuate nucleus and part of the VMH, other brain regions should also be shown including lateral hypothalamus and DMH, because which also play important roles in the regulations of energy and glucose metabolism. They did show other brain regions in the supplementary figure 1, however, they performed the experiments in fast but not in fed conditions (Figure 1a-c). The experiments should be performed in same conditions, otherwise the conclusion for the Figure 1 was not convincing.

In Figure 1g and i, fasting induced c-fos in the ARC in normal mice. What about the effects of fast on c-fos expression in T1D mice? which is necessary to compare to fed T1D mice.

For the Fig. 1j, please explain why fast did not decrease blood glucose. Meanwhile, the authors should tone down this sentence "T1D and fasting share a common mechanism in which heightened activation of LepArC neurons owing to reduced leptin action causes augmented counter-regulatory responses", because T1D and fasting are two different metabolic conditions which would cause various hormonal and metabolic changes.

For the Fig. 1l and m, an important control is missing regarding the c-fos positive AgRP neurons in non-T1D animals. Also, what are the physiological roles of these activated AgRP neurons in T1D?

Continued for the Figure 1, what about leptin control of food intake and body weight in T1D? which should be performed accompanied with glucose detection.

Figure 2:

For Figure 2a-f, there are also leptin receptor expressing neurons around the ARC including the VMH next to the ARC, however, there was no or little viral expressions in the VMH. It was surprising because it is impossible to limit viral vectors in the ARC without vector spreading to other brain regions. In the methods, the authors mentioned that "4 local injections with 2 each side, 200nl/each side), which was huge to infect the medio-basal hypothalamus including VMH and DMH, but not only the ARC.

For Fig. 2g, another important control was missing regarding the control vector mice with CNO treatment, because CNO deprivation might bring non-specific effects on blood glucose (Howes OD et al., *J Clin Psychiatry*, 2013). The label in the text "toward this...(Fig. 2g)" was not consistent with the Fig. 2i.

For the Fig. 2j-i, there are interesting experiments, while I was confused. The Figure 1 demonstrated that GABAergic neurons were activated in T1D animals, why was the NachBac approach used to activate these neurons in T1D animal? These neurons are already activated in T1D. Meanwhile, it is necessary to apply higher dose of leptin to the vGAT-Cre mice, possibly reaching the leptin effect on AgRP-Cre mice because AgRP neurons are just subpopulation of GABAergic neurons. Otherwise it is difficult to draw the conclusion. It also is required to monitor food intake and body weight accompanied with glucose changes.

For Figure 3, control vector transduced animals should be included.

For Figure 5d regarding the AgRP-DTR::vgat-Cre mice, my understanding was that they crossed AgRP-Cre with iDTR mice to get heterozygous AgRP-DTR mice, which was subsequently crossed

with Vgat-Cre mice. If so, vGat neurons might also express DTR. Please clarify the offsprings used in this study, and provide detailed information regarding how to screen the animals.

Responses to reviewers' comments

We would like to thank all reviewers for their insightful comments. The following is our point-to-point responses to these comments.

Reviewer #1 (Remarks to the Author):

- 1) *The work shows that ARH GABAergic neurons mediate the effect of leptin, and that the effect does not require AgRP neurons. The work also suggests that nutrient sensing is defective in ARH GABAergic neurons, although I was not strongly convinced of this data, and felt it would be worth considering the effect of insulin in this part of the work.*

Response: We thank the reviewer for the raising the issue on nutrient sensing needing in consideration of insulin. We want to emphasize that the evidence we provided to support leptin action in nutrient sensing is compelling. Our current data showed **1)** in T1D, Arc LepR neurons are defective in changes of c-Fos expression in response to fasting with a drastic change in glucose levels (Fig. 6a-d), which is in contrast to a more drastic change in c-Fos with much less changes in glucose levels in controls (Fig. 6f), suggesting a defective response to glucose in vivo; **2)** our in vitro slice recording showed that, Arc LepR neurons in T1D failed to respond to glucopenic 2-DG (Fig. 6h), which can be reversed by leptin, pointing to direct evidence that those neurons in T1D are defective in glucose sensing (Fig. 6i); **3)** The in vitro data on 2-DG and leptin is directly supported by in vivo data showing 2-DG reverses leptin effects on reducing glucose in T1D (Fig. 6k and 6l); **4)** The effect of leptin on reducing T1D glucose is achieved by its inhibitory action (Fig. 1d, Fig. 3d and Fig. 4b), which is consistent with the 2-DG action on reversing leptin action on c-Fos expression in T1D Arc neurons (Fig. 6m and 6n); **5)** consistent with a key role in energy sensing of AMPK, Arc LepR neurons exhibit AMPK activation in T1D, which is reversed by icv leptin; and **6)** importantly, our new data show that the reversal of leptin effects on reducing T1D glucose by 2-DG is associated with AMPK activation in Arc LepR neurons, directly linking leptin action and 2-DG with AMPK activity within Arc LepR neurons in regulating T1D glucose. These collective data convincingly suggest that Arc LepR neurons in T1D, an extremely low leptin state, is defective in glucose sensing, which can be reversed by leptin treatment.

The reviewer's point is that it would be more convincing if our data take the insulin action into consideration. However, we have to respectfully point out that, despite considerable amount data on direct insulin action on these neurons, insulin action is not relevant to the current investigation: **a)** the goal and the strength of the current study is to investigate the effect of brain leptin action on reducing T1D glucose in an insulin independent manner, and the data on insulin may not provide additional insights on the effect of icv leptin on restoring T1D glucose and will likely cause unnecessary distraction to the focus of the current study; **b)** it is known that brain leptin infusion is effective in reducing T1D hyperglycemia while i.c.v. insulin is not capable of doing it (1, 2), suggesting that brain leptin action on glucose in T1D is distinct from insulin; **c)** consistently, brain insulin and leptin action in the Arc are suggested to be mediated by distinct groups of neurons (3), and brain insulin action is independent of leptin action (4) and vice versa (5). Thus, although we agree with the reviewer that insulin action in the brain is important but we think for this particular study, data on insulin action on glucose sensing is not necessary and will not add more weight to our overall conclusions regarding brain leptin action on reducing T1D glucose in an insulin independent manner.

- 2) *The studies appear well conducted, and the data are well explained. The results are believable. The discussion is appropriate with the caveat that the clinical data needs to be discussed, and they need to reiterate in the discussion that this therapy was tried in humans and did not produce clinically meaningful changes in glucose levels.*

Response: We thank the reviewer for this important point. We have added the following point to Discussion: **Notably, recent human clinical trial with a combination of insulin and leptin treatment of T1D shows that leptin fails to provide a significant benefit. Although this failed clinical trial, which is in contrast to the demonstrated leptin effect in rodents, may be due to many reasons, including differences between humans and rodents, one key difference in human T1D patients is that these patients are always on insulin treatment, and the leptin level is only reduced 50% (6-8), compared to non-T1D humans, this is likely due to the fact that insulin is capable of rapidly increasing leptin levels (9, 10). In contrast, in rodent T1D models, including those used in this study, mice were not received any insulin treatment and as a result, leptin levels were extremely low, compared to controls (9, 11-13). It is thus conceivable that additional leptin treatment on top of this mildly reduced leptin condition in human T1D may not be able to provide additional beneficial effects. In other words, the beneficial effect of brain leptin action may have been reflected by the effects of insulin treatments. Further studies are warranted to specifically address this possibility.**

- 3) *Edit to methods section: I did not see any fiber optic work, it may have been an earlier version. "Stereotaxic surgeries to deliver viral constructs and for optical fiber implantation were performed as previously described 3."*

Response: We are sorry for our neglect and this description has been removed in the revised version.

- 4) *Ref 3 is: Fujikawa, T., Chuang, J. C., Sakata, I., Ramadori, G. & Coppari, R. Leptin therapy improves insulin-deficient type 1 diabetes by CNS-dependent mechanisms in mice. Proceedings of the National Academy of Sciences of the United States of America 107, 17391-17396, doi:10.1073/pnas.1008025107 (2010). I assume this is incorrect reference?*

Response: After careful examination, we think this is a correct reference, as this study represents one of the first studies demonstrating leptin action in the brain is sufficient to reduce STZ-T1D in an insulin-dependent manner. Although several previous studies have suggest this, the conclusion in those studies is not explicit on the nature of insulin independency (14). If the reviewer suggests it is necessary to include those citations, we will be happy to do so.

Reviewer #2 (Remarks to the Author):

I read this with interest and there is much of merit in this manuscript.

In particular, there are a lot of elegant studies which appeared to have been performed with great technical capability and the number of different experimental approaches (some of them beautiful) have been employed.

The underlying premise is that a model of T1D (mostly streptozotocin-induced diabetes with one study also done in NOD mice) is characterised by low leptin levels which leads to activation of key neuronal populations within the arcuate nucleus.

Response: We thank the appreciation of the reviewer on our work.

- 1) Although a “presentational”, I am concerned by the repeated use of T1D throughout the manuscript. This isn’t T1D but a model of untreated insulopenic diabetes. Even in the NOD studies where immune mediated damage, mice are not treated with insulin replacement. I think they need to be clearer about this. This may seem pedantic but this is very different from T1D and of course this is the reason for low leptin. Parenthetically, were circulating leptin levels measured?

Response: We thank the reviewer for pointing out this important point. To clarify this and avoid potential confusion to readers, we have made a special explanation on the reference of T1D in the discussion section, stating that **“It is important to note that the term of T1D referred here is specifically for a state of untreated insulopenic diabetes induced by STZ, which is distinct from the typical type 1 diabetes condition in humans with insulin treatment and largely controlled glucose levels.”**

We didn’t measure leptin levels as the extremely low leptin levels in STZ-T1D have been well demonstrated by numerous other studies (10-12, 15) and it is now well accepted that the low leptin level in insulin deficiency causes hyperphagia and increased counter-regulatory responses, which drive hyperglycemia (10, 15-17).

- 2) *David Coppari showed some years ago that leptin acts via hypothalamic GABAergic neurones to lower blood glucose (Cell Metabolism 2013 doi: 10.1016/j.cmet.2013.08.004) so the novelty here appears to be narrowing this down to arcuate and nicely showing with DREADD that leptin inhibits target cells. Their findings are also consistent with Lowell’s arcuate “census” n11.Trh/Cxcl12 neurons which were (non-AGRP) GABAergic leptin responsive population. I think both of these papers are worth incorporating*

Response: The Coppari paper has been cited as Ref. 5 and the Lowell paper has been cited as Ref. 13 in the manuscript. To specifically emphasize on the point of non-AgRP neurons, the Lowell paper has also been newly cited in the Discussion on the potential role of non-AgRP neurons in glucose homeostasis.

- 3) *Scientifically, the approach is a little confusing as they have flipped from using LepR-cre to GLUT-cre. This makes this scientifically heterogenous as some of the studies are looking at arcuate leptin receptor containing neurones*

Response: We thank the reviewer for the point on using different Cre mice to target Arc neurons. We agree with the reviewer that it would be ideal to use LepR-Cre for all experiments in this studies. In the current study, we used LepR-Ires-Cre to target Arc neurons in Fig. 2a-h and used Vgat-Cre/AgRP-Cre target Arc neurons in all other studies with vector delivery. One of the major reason for this change is the feasibility of using LepR-Cre to specifically target Arc neurons. As rightfully raised by Reviewer 4 below in his/her comments on the data presented Fig. 2, since LepR-Ires-Cre also targets VMH neurons, and given the close proximity between the Arc and VMH, it is difficult to achieve Arc-specific delivery. It has taken us a great deal of

efforts in screening out successful delivery from many mice to accomplish the experiments with LepR-Ires-Cre presented in Fig. 2a-2h. We then re-assessed the project and decided to switch to using Vgat-Cre mice because the regions surrounding Arc are mainly glutamatergic neurons, which makes it much easier and feasible to achieve Arc-specific delivery. We have successfully used this Vgat-Cre to target Arc GABAergic neurons (18, 19). However, switching to Vgat-Cre will not affect the main conclusion of this study because **a)** Previous studies suggest that GABAergic neurons (i.e. Vgat-Cre) mediate the leptin action in both body weight and reducing T1D glucose (16), suggesting that the LepR neurons that mediate the leptin action on glucose is within the GABAergic neuron group. Positive results from the current studies will point to a role of GABAergic LepR neurons in the Arc; and **b)** most of our studies using Vgat-Cre with viral delivery are performed with icv leptin pharmacology (Fig. 2i-l, and Fig. 4), which directly relates to leptin action via its receptors.

4) *Corticosterone and/or glucagon data are shown but no clear whether both were measured in all studies or whether only the hormone shown was measured and why selected?*

Response: It is generally accepted that brain leptin action (icv infusion) on reducing T1D hyperglycemia is mediated by suppressing counter-regulatory responses (6). The known counter-regulatory responses include glucagon, HPA axis and sympathetic nerve output (norepinephrine). A role for beta-adrenergic receptor-mediated sympathetic nerve output has been ruled out (16, 20). Heightened glucagon action has been suggested to be essential (21), but was later disputed on whether it depends on insulin action (22). The HPA axis has also been suggested to be the mediator (23, 24), but also was later disputed (25, 26). It is thought that multiple counter-regulator responses are involved and suppression of one of them may not be able to be sufficient to explain the leptin effects (6, 25).

Based on the above information, we choose to measure glucagon and/or corticosterone but not epinephrine/norepinephrine to document relevant changes of counter-regulatory responses. However, it is important to point out that the purpose of our measurements of glucagon and corticosterone is not to suggest the underlying mechanism responsible for the changes in glucose, but rather to document the association between counter-regulatory responses and glucose changes and to confirm brain leptin or leptin-mimicking or anti-leptin actions. Therefore, since demonstrations of glucagon, corticosterone or both equally document the changes in counter-regulatory responses, it will serve the same purpose to show either one or both of them.

Nonetheless, in the current studies, we provided both glucagon and corticosterone measurements in Fig. 3, Fig. 4 and Fig. 5, and our new data on glucagon has been added to Fig. 2 in this revision.

5) *Although I accept that some of their other data suggest that AGRP neurons not involved in this action of leptin, I am unconvinced by the DTR ablation studies (Fig 5 etc). We know from other studies that there is compensation with early life ablation of AGRP neurons resulting in no real phenotype (whereas adult ablation is rapidly fatal). My bias is to remove this study from manuscript*

Response: We thank the reviewer for raising the important issue on AgRP neuron lesion. However, we have to respectfully argue against the reviewer assessment of the use of the data. We have the following points to support our view. **a)** Our conclusion on the role of AgRP neurons is not from the experiment with AgRP neuron lesion alone, which is also directly confirmed by our

other results from chronic AgRP neuron inhibition. Our chronic AgRP inhibition model is achieved with Kir2.1 expression in adult mice, which has no concerns of developmental compensations. **b)** Several published studies suggest that starvation caused by DTX-induced AgRP killing might be due to a rapid disturbance or potential secondary effects. Previous results on adult AgRP killing demonstrate that starvation is due to loss of GABA release(27). However, a recent publication from Wu and Palmiter suggests that disruption of GABA release from adult AgRP neurons causes little effects on feeding (28). In addition, earlier results from Palmiter on the starvation effect from adult AgRP killing can be rescued by obesity (29), suggesting a non-specific effect rather than a specific neurocircuit defect, which would otherwise cause starvation irrespective of obesity status. **c)** Adult killing of AgRP neurons is known to cause gliosis (30) and potential other issues and moreover, physical structural damage in nearby structure is conceivable to cause non-specific effects. **d)** Several studies published from Palmiter and other labs using the AgRP neonatal ablation model to draw conclusion that the physiological processes under study are independent of AgRP neuron function (31, 32). **e)** Our previous study using Kir2.1 to chronic inhibit AgRP neurons shows no effects on body weight (18). **f)** Our preliminary study in which we kill AgRP neurons in adult mice with direct diphtheria toxin expression (without a need of DTX injection) through delivery of AAV-Flex-DTX to AgRP-Cre mice shows no changes in body weight.

With this collective evidence, we hope that the reviewer agrees with our assessment that the data from AgRP lesion presented here can at least serve as additional supportive data for the results on our other model with chronic AgRP inhibition.

It is important to point out that our data do not support that AgRP neurons have no role in mediating the leptin action and instead our data support an importance of AgRP neurons, but also support the importance of non-AgRP neurons, in mediating the leptin action.

6) *“The leptin inhibitory action on LepRArc neurons were reversed by 2-DG (Fig. 6g), suggesting that the leptin inhibition requires nutrient supply” If anything, figures 6 g,h and I show that leptin exerts its inhibitory effect regardless of glucose manipulations?*

Response: We agree with the reviewer that Fig. 6 g, h and I show that leptin exerts inhibitory effects; however, our experimental condition was maintained in constant 1 mM bath glucose throughout, with no manipulations in glucose levels. It is clear that in control neurons, all the inhibitory action of leptin is reversed by 2-DG, which essential signals low glucose. In contrast, for T1D neurons, 2-DG has no obvious effects but gains effect with leptin, suggesting T1D neurons lose nutrient sensing without leptin action.

7) *Should the label in fig 2 panel c in red read mCherry not Gq DREADD?*

Response: We feel sorry for our neglect and this mistake has been corrected in the revised version.

8) *I am unconvinced by the 2DG studies. To my eye, figure 6j just shows a ceiling effect ie glucose levels and presumably stress/ counterregulatory hormones are already very high and little scope for further elevation with glucopenic challenge?*

Response: We have to point out that Fig. 6k provides compelling evidence that icv 2DG reverses leptin action on reducing glucose. Specifically for Fig. 6j, we agree with the reviewer that we couldn't rule out there might be a ceiling effect. However, we know that mouse glucose levels can be >35mM as we frequently observe glucose levels >35mM (or exceeding detect limit

of glucometer) in T1D mice, if 2DG hyperglycemia and T1D hyperglycemia are caused by parallel pathways, i.p. 2DG injection should increase the glucose level in T1D mice. In this study, we reason that providing this set of data will add more evidence to our conclusion. However, to acknowledge the possibility of a potential ceiling effect, we have added “**However, it couldn’t be ruled out that no increase in glucose in response to 2DG might be due to a ceiling effect**”. However, if the reviewer think that it is not wise to include this set of data, we could remove it. Nonetheless, we believe that our data presented in Fig. 6k and other associated data on glucagon and c-Fos convincingly demonstrate that 2DG reverses the leptin action.

9) *“LepRArc neurons in T1D were not sensitive to 2-DG (Fig. 6h), consistent with the notion that these neurons lose nutrient sensing”. These are neurons that have been bathed in vivo in high levels of glucose but then studied ex vivo at 1 mM. My feeling is that they will already be operating under an environment they perceive as glucopenic so not surprising that no further effect of 2DG. It is difficult to know whether this is plausible as data are just presented as bar graphs normalized to 1 mM glucose conditions (designated “100%”).*

Response: We thank the reviewer for this important consideration. We want to point out: **a)** all recorded neurons have reached a steady state in activity at 1mM before 2DG experiments; **b)** all recorded neurons, regardless of controls and T1D, have been bathed in high glucose levels during brain slice sectioning and subsequent incubation period, which is required to obtain high-quality neurons for in vitro recording; and **c)** our results on greatly increased c-Fos expression in these neurons in T1D (very high glucose levels) in a similar fashion to fasting (low glucose), and reversal of T1D c-Fos by leptin, suggest that even though they are bathed *in vivo* in high glucose (i.e. T1D), these neurons are not be able to sense it but instead are in a glucose-deprived state.

10) *Studies shown in figures 6a b e are surely largely a scientific iteration of those shown in fig 1?*

Response: We agree that both Fig 6a/b/e and Fig 1 contain c-Fos data. However, the purpose of these data is different. In Fig1, the purpose is to show that both fasting and T1D causes c-Fos in Arc LepR neurons, which can be reversed by leptin. In contrast, the purpose of Fig. 6 is on glucose sensing, in which we demonstrated that despite drastic changes in glucose levels, no change in c-Fos was observed in T1D, while with slight changes in glucose levels, drastic changes in c-Fos was observed in controls, demonstrating loss of glucose sensing in T1D.

Reviewer #3 (Remarks to the Author):

This manuscript describes the results for experiments designed to investigate the role of leptin receptor positive arcuate neurons in T1D-mediated hyperglycemia. The experiments are well designed, the Methods utilised appropriate, figures are of excellent quality and the overall data presentation and interpretation is very good. The findings and conclusions presented herein are very interesting, novel and should have a significant impact for those interested in understanding the aetiology of diabetes and the complications associated with chronic and severe hyperglycemia. There are some minor errors and

typographical mistakes to be corrected. In addition, there are some comments below that need addressing to clarify certain issues.

Response: We thank the reviewer's appreciation of our work.

General comments:

- 1) *. In the Introduction, define what the CRR is for readers and what constitutes the response. The CRR (which often becomes defective in many individuals with T1D) is normally associated with hypoglycemia excursions than chronic hyperglycemia and this could be made clearer in the text in relation to T1D and glycemic variability.*

Response: We thank the reviewer for this important background. We have added these sentences to the Introduction. **"It is generally accepted that brain leptin action on reducing T1D hyperglycemia is mediated by suppressing counter-regulatory responses (CRR) (6). The CRR includes glucagon, HPA axis and sympathetic nerve output (epinephrine and norepinephrine), which normally represents hypoglycemic responses for glucose homeostasis but is aberrantly activated in T1D. For brain leptin action on reducing glucose in T1D, a role for the sympathetic nerve output has been ruled out (16, 20). Heightened glucagon action has been suggested to be essential (21), which was later suggested to be dependent on insulin action (22). The HPA axis has also been suggested to be the mediator (23, 24), but also was later suggested not to be a sole mediator (25, 26).**

- 2) *The hormones associated with the CRR are not all represented in this manuscript. Although data for glucagon and corticosterone are presented in many figures, no mention is made of epinephrine, a key component of the CRR. Did the authors monitor the plasma levels of epinephrine? If so can they present these data and if not, can they state why they did not?*

Response: It is reminded that the goal of the current study is to identify brain neurons and their mechanisms underlying leptin action on T1D glucose. As the role of glucagon and the HPA axis in mediating brain leptin action on reducing T1D diabetes has been under intense debate and controversy (6, 22, 23, 33), we want to point out that the purpose of our measurements of glucagon and corticosterone were not to demonstrate the underlying mechanism responsible for the changes in glucose, but rather to document the association between counter-regulatory responses and glucose changes and to confirm brain leptin or leptin-mimicking or anti-leptin actions. Therefore, since demonstrations of glucagon, corticosterone or both equally document the changes in counter-regulatory responses, it will serve the same purpose to show either one or both of hormones.

As mentioned above, since a contribution of the beta receptor-mediated sympathetic nerve output to the leptin action has been ruled (16, 20), we didn't measure epinephrine in this study.

- 3) *The main mouse model of T1D that is used here is the STZ-treated animal. However, this leaves the animal with little insulin. Do the authors consider low residual insulin as having enhanced action in the presence of leptin – other studies suggest not – but could this issue be alluded to in Discussion? Also, in humans T1D would be treated by insulin therapy to reduce BG. What is the evidence that leptin would be a useful adjunct therapy under these conditions? Please add some discussion of this point.*

Response: This comment is similar to comment 2 from Reviewer 1. In this study, as we previously showed, the leptin action is independent of insulin action. As the reviewer points out, humans T1D normally receives insulin treatments, and as insulin is capable of rapidly increasing leptin levels, the beneficial effect of insulin treatment is likely contributed by brain leptin action. We have added the following to the Discussion, which we think will extend the current finding to human T1D treatments.

“Notably, recent human clinical trial with a combination of insulin and leptin treatment of T1D shows that leptin fails to provide a significant benefit. Although this failed clinical trial, which is in contrast to the demonstrated leptin effect in rodents, may be due to many reasons, including differences between humans and rodents, one key difference in human T1D patients is that these patients are always on insulin treatment, and the leptin level is only reduced 50% (6-8), compared to non-T1D humans as insulin is capable of rapidly increasing leptin levels (9, 10). In contrast, in rodent T1D models, including those used in this study, mice were not received any insulin treatment and as a result, leptin levels were extremely low, compared to controls (9, 11-13). It is thus conceivable that additional leptin treatment on top of this mildly reduced leptin condition in human T1D may not be able to provide additional beneficial effects. In other words, the beneficial effect of brain leptin action may have been reflected by the effects of insulin treatments (34). Further studies are warranted to specifically address this possibility.”

- 4) *Is the chronic activation of these arcuate GABA neurons simply down to the reduced plasma leptin, or will other factors/hormones associated with T1D state also contribute – such as relative lack of insulin – as insulin known to act on arcuate neurons to alter electrical activity and modify glucose homeostasis (e.g. acting on AgRP neurons).*

Response: Yes, we think that the activation of these neurons is due to reduced leptin because T1D shows extremely low leptin levels and leptin administration greatly reduces c-Fos expression (Fig. 1). It is unknown whether there are other factors contributing to it. However, the brain insulin action is unlikely to contribute. It is known that, while brain leptin infusion is effective in reducing T1D hyperglycemia, i.c.v. insulin is not capable of doing it (1, 2), suggesting that brain leptin action on glucose in T1D is distinct from insulin. Consistently, brain insulin and leptin action in the Arc are suggested to be mediated by distinct groups of neurons (3), and brain insulin action is independent of leptin action (4) and vice versa (5). However, since leptin levels can be rapidly increased with insulin treatment in T1D (10), part of insulin action on reducing glucose is likely contributed by brain leptin action on glucose.

- 5) *Electrophysiological characterization showing primary data is presented in a Supplemental Figure where the authors show an exemplar whole-cell recording from one LepR neuron and relative numbers for excitability changes. The concentrations of glucose used in these experiments need to be explained in light of the plasma glucose levels presented in the other figures – ranging from ~6 to 30 mM. Why were 1 mM and 10 mM glucose used for these experiments. Also, readers should be reminded that brain glucose levels are much lower than*

plasma glucose and 1 mM would be equivalent to hypoglycemia whereas 10 mM is much higher than central neurons are likely to experience (~4.5 mM is usually the upper limit in severe hyperglycemia according to published figures).

Response: We thank the reviewer for raising this important point. The main figures with glucose levels ranging from 2-30mM show changes in glucose levels during various physiological states (T1D, fed, fasting, leptin treatments, with Arc neuron activation or inhibition).

Specifically for recording experiments on in vitro brain slices, the neurons are situated at a very different environment. First, with a standard protocol of brain sectioning and incubation, the brain sections are incubated with a very high glucose concentration around 10mM glucose, which gives a very good quality of neurons for recording. Second, these neurons in brain slices are without functional blood vessels or ventricles, and therefore may be difficult to access glucose or other factors in the incubation buffer. Nevertheless, we used 1mM as a starting glucose concentration to mimicking a low glucose level, for example in Fig. 6, in which neuron firing rate is presumably higher, aiming for a bigger window to observe leptin inhibitory action. In Supplementary Fig. 7, the starting glucose level is 1 mM, and then increased to 10 mM, a very high glucose levels, as the Reviewer rightfully pointed out, grossly mimicking brain glucose levels in T1D, to see the effect of high glucose on neuron activity change.

To clarify the selection of these concentrations, we've added clarification sentences in the method section.

6) *In Figure 6 a-f, comparing c-fos expression in the fasted-fed transition in non-diabetic vs T1D mice. Why is BG reduced to such a large extent on fasting with a high proportion of Arc c-Fos+ neurons – which as you have shown clearly correlate with high plasma glucose? Presumably these T1D neurons are not able to undergo hyperpolarisation/reduced firing under fasting conditions (as you have argued – loss of ability to nutrient sense) – so what is causing BG to decline here? In your model you show 2DG + leptin increases firing of ARC LepR+ neurons suggesting nutrient sensing disturbance and reduced glucose metabolism giving rise to excitation. So in fasted state with diminished glucose metabolism (no leptin added?) these neurons will be excited as you show – so expect a higher BG associated with increased levels of glucagon and cort? Please explain?*

Response: We thank the reviewer for raising this interesting point. In T1D, the Arc neurons are activated and the counter-regulatory responses (CCR) are also activated. Due to chronic activation of CCR, there will presumably be little glycogen reserve in the liver and the gluconeogenic pathways will likely to be maximally activated. Under this condition, the animal has to increase feeding to provide glucose sources, in other words, feeding has become a key energy source. Thus, these mice are hyperphagic and very sensitive to fasting, and with a relatively short period of fasting, their glucose drops rapidly, as shown in Fig.6f. It is also well known that human T1D patients require frequent meals. We hope that the reviewer agrees with our reasoning.

7) *Figure 7. Activation of AMPK associated with increased BG and in T1D, LepR+ neurons exhibit high p-AMPK. Leptin decreases p-AMPK in the T1D model – so my question is does addition of 2DG act to increase p-AMPK in Arc LepR+ neurons in leptin-treated T1D mice in association with raised BG?*

Response: We have provided these data in the revised version of Fig. 7a and Fig. 7b.

- 8) *No mention of how the effects of central leptin in the T1D model are transmitted to peripheral tissues to improve BG levels. Discuss role of central leptin pathway to ANS to alter glucose uptake in muscle and BAT as well as diminished hepatic glucose output? How might this be tested – use of a ganglion-blocker?*

Response: As mentioned above, the mechanism on how the pathways (mainly CRR) to peripheral tissues mediate the leptin action have received intense attention and generated debates on whether glucagon or HPA axis alone mediates the effects (6, 21-26). As mice with deficiency in all 3 beta-Adrenergic receptors have normal response to leptin in reducing T1D glucose, a role for sympathetic nerve output is ruled out (16, 20). However, as this study is focused on brain mechanism of leptin action, experiments on specific pathways from brain to the peripheral tissues, although important (as evidenced from intense investigations above), are not the focus of the current study.

As requested by Reviewer 2, we have provided this background information on debates regarding the pathways from the brain to peripheral tissues in Introduction.

Minor Comments:

1. *Figure 1 legend: ANOVA?*

Response: This has been corrected.

2. *Page 6 L11-12: “..expression was apparent in both AgRP and non-AgRP neurons...? Please clarify what is meant here.*

Response: Sorry for the confusing statement. This has been corrected as “expression was present in both AgRP and nearby non-AgRP neurons.

3. *Page 8 L15-17 & Fig3a,b: (a) -AgRP (b) – Vgat The text has (a) as Vgat and (b) as AgRP*

Response: Sorry for the mistake. This has been corrected in the revised version.

4. *P9 L18-19 Fig 4d,e: In text cort levels (4d) and glucagon (4e) but in legend and figure glucagon is (d) and Cort is (e)*

Response: Sorry for the mistake. This has been corrected in the revised version.

5. *Page 10 L5-6 Fig 5b: Blood glucose levels for saline vs DTX treated T1D animals in text and figure legend but shown as saline vs leptin in figure. Please clarify.*

Response: Sorry for the mistake. For blood glucose levels it should be labelled as controls and DTX treated groups (both treated with STZ) in both text and figure legend. This has been corrected in the revised version.

6. *Supplemental Fig 5: Representative WCR – please give an indication of Vm for these traces and comment on any changes in this parameter with altered glucose and/or by leptin?*

Response: We have updated Supplemental Fig 5 with Vm changes.

7. *Fig 7a middle image: The arrows are not aligned across the panels and there is no description relating to the meaning of the arrows in legend or text*

Response: The arrows have been aligned and description of the arrow has been provided in the figure legend and text.

8. *P16 L1-2 : This sentence needs addressing – ... in face with ...? Unclear what is meant here. Also next sentence – LepRArc neurons lose responses...? Line 5 further*

Response: We have revised these sentences to avoid confusion.

Reviewer #4 (Remarks to the Author):

- 1) *The neuron populations for leptin control of glucose in normal and diabetic animals have already revealed by previous studies (i.e. Fujikawa T et al., Cell Metabolism, 2013) including GABAergic and POMC neurons in the hypothalamic arcuate nucleus. The authors reported that leptin receptor expressing neurons in T1D animals were activated, which might be attributable to decreased leptin. In this study, plasma leptin should be detected in the experimental conditions. Meanwhile, it is also important to examine the mRNA and/or protein expressions of leptin receptors in the identified neuron populations.*

Response: As discussed in the response to Reviewer 1, leptin levels in the STZ-induced T1D mouse model are well-established to be extremely low (9-12, 17). It is also well-accepted that insulin treatment causes a rapid increase in leptin (9, 10). In this case, data on leptin levels may not necessarily provide any additional information.

As for leptin receptor expression, we also think this specific information will not be able add more weight to our conclusion. Given the well accepted extremely low leptin in STZ-induced T1D, the results on leptin receptor expression, whether it will be increased, reduced or no change, will not prove or disapprove the over conclusion from this study on leptin inhibitory action on Arc neurons through the mediation of nutrient sensing reduces counter-regulatory responses and T1D glucose.

- 2) *In Figure 1a-c, in addition to the arcuate nucleus and part of the VMH, other brain regions should also be shown including lateral hypothalamus and DMH, because which also play important roles in the regulations of energy and glucose metabolism. They did show other brain regions in the supplementary figure 1, however, they performed the experiments in fast but not in fed conditions (Figure 1a-c). The experiments should be performed in same conditions, otherwise the conclusion for the Figure 1 was not convincing.*

Response: We feel sorry for the way we described Supplementary Fig. 1, which causes the confusion. Our mouse models used here were either fasting or fed T1D. Supplementary Fig. 1c-

g was from fed T1D, the same mice shown in Fig. 1a-d. We have revised the Figure legend of Supplementary Fig. 1 in the revised version.

3) *In Figure 1g and i, fasting induced c-fos in the ARC in normal mice. What about the effects of fast on c-fos expression in T1D mice? which is necessary to compare to fed T1D mice.*

Response: We thank the reviewer for this intriguing question. As a matter of fact, we performed the exact experiment and results are shown as Fig. 6c, 6d, 6e and 6f. The results demonstrate that c-Fos+ LepR neuron number in the Arc was not different between fed and fasting in T1D (Fig. 6e), although their glucose levels exhibited dramatic difference (Fig. 6f), which provides compelling evidence that, in contrast to control (drastic change in c-Fos with small changes in blood glucose), these neurons in T1D are not capable of sensing glucose changes.

4) *For the Fig. 1j, please explain why fast did not decrease blood glucose. Meanwhile, the authors should tone down this sentence "T1D and fasting share a common mechanism in which heightened activation of LepArc neurons owing to reduced leptin action causes augmented counter-regulatory responses", because T1D and fasting are two different metabolic conditions which would cause various hormonal and metabolic changes.*

Response: We feel sorry that we failed to make it clear that fasting described in Fig. 1j is a 8hour fasting (from early morning to late afternoon), which didn't cause a significant drop in glucose in controls, but caused a dramatic rapid drop in T1D. We have added this important information in the revised version.

We agree with reviewers that T1D and fasting are 2 different physiological states. However, when we state that T1D and fasting share a common mechanism, we specified specific common changes including Arc neuron activity, leptin levels, counter-regulatory hormones etc. We have published a review article with specific discussion on this point (35).

5) *For the Fig. 1l and m, an important control is missing regarding the c-fos positive AgRP neurons in non-T1D animals. Also, what are the physiological roles of these activated AgRP neurons in T1D?*

Response: We have added the pictures and associated data, shown below, in the revised version as part of Fig. 1l.

6) *Continued for the Figure 1, what about leptin control of food intake and body weight in T1D? which should be performed accompanied with glucose detection.*

Response: The effect of feeding and body weight in T1D has been examined in our previous studies (20), as well as in numerous other studies (11, 16, 17, 36, 37), and thus, duplicating the same set of data will not add further information that will strengthen our conclusion. In particular, for studies in Fig 1 on fasting conditions, it is not feasible to obtain data on feeding per se.

7) *For Figure 2a-f, there are also leptin receptor expressing neurons around the ARC including the VMH next to the ARC, however, there was no or little viral expressions in the VMH. It was surprising because it is impossible to limit viral vectors in the ARC without vector spreading to other brain regions. In the methods, the authors mentioned that "4 local injections with 2 each*

side, 200nl/each side), which was huge to infect the medio-basal hypothalamus including VMH and DMH, but not only the ARC.

Response: We thank the reviewer for pointing out the difficulty in using LepR-Ires-Cre to target Arc neurons. In the current study, we used LepR-Ires-Cre to target Arc neurons only in Fig. 2a-h and changed to use Vgat-Cre/AgRP-Cre to target Arc neurons in all other studies with vector delivery. One of the major reason for this change is the feasibility of using LepR-Ires-Cre to specifically target Arc neurons. As rightfully raised by the Reviewer 4, since LepR-Ires-Cre also targets VMH neurons, and given the close proximity between the Arc and VMH, it is difficult to achieve Arc-specific delivery. It has taken us a great deal of efforts in screening out successful delivery from many mice to accomplish the experiments with LepR-Ires-Cre presented in Fig. 2a-2h. We then re-assessed the project and decided to switch to using Vgat-Cre mice because the regions surrounding Arc are mainly glutamatergic neurons, which makes it much easier and feasible to achieve Arc-specific delivery. However, switching to Vgat-Cre will not affect the main conclusion of this study because **a)** Previous studies suggest that GABAergic neurons (i.e. Vgat-Cre) mediate the function of leptin and leptin receptors in both body weight and reducing T1D glucose, suggesting that the LepR neurons the mediate the leptin action on glucose in within the GABAergic neuron group. Positive results from the current studies will point to a role of GABAergic LepR neurons in the Arc; and **b)** most of our studies using Vgat-Cre with viral delivery are performed with icv leptin pharmacology (Fig. 2i-n, and Fig. 4), which directly relates to leptin action via its receptors.

To clarify this point, we have provided specific justification of switching from LepR-Ires-Cre to Vgat-Cre when we described the results in the manuscript.

The description in the method is for Vgat-Cre injections but not for LepR-Ires-Cre and we apologize for this mistake. We have provided separate descriptions for both injections in the revised version.

On a different note, with enough time/experience, it is possible to achieve Arc specific delivery, sparing VMH, even with Vglut2-Cre, which shows abundant expression of Cre-expression the VMH and little expression in the Arc, it has been reported that injections of 5nl Cre-dependent viral particles achieve specific expression in the Arc but not in the VMH (38).

- 8) *For Fig. 2g, another important control was missing regarding the control vector mice with CNO treatment, because CNO deprivative might bring non-specific effects on blood glucose (Howes OD et al., J Clin Psychiatry, 2013). The label in the text "toward this....(Fig. 2g)" was not consistent with the Fig. 2i.*

Response: As requested, we have performed a new experiment to test the potential effect of CNO in a new cohort of mice (shown below), and presented as Supplementary Fig. 3 in the revised version.

- 9) *For the Fig. 2j-i, there are interesting experiments, while I was confused. The Figure 1 demonstrated that GABAergic neurons were activated in T1D animals, why was the NachBac approach used to activate these neurons in T1D animal? These neurons are already activated in T1D. Meanwhile, it is necessary to apply higher dose of leptin to the vGAT-Cre mice, possibly reaching the leptin effect on AgRP-Cre mice because AgRP neurons are just subpopulation of*

GABAergic neurons. Otherwise it is difficult to draw the conclusion. It also is required to monitor food intake and body weight accompanied with glucose changes.

Response: We thank the reviewer for this stimulating question. The purpose of this experiment is to test whether leptin-induced inhibition of AgRP or Vgat-Cre neurons is necessary for the leptin effect on glucose. The expression of NachBac will cause neuron activation as shown in Fig. 2i, and this NachBac-mediated activation will block the inhibitory effect of leptin (Fig. 2k-2l). Thus the NachBac-mediated activation can be used to test whether specific leptin inhibition of Vgat-Cre or AgRP neurons is necessary for leptin action on glucose. Since we have evidence that leptin has successfully engaged its signaling, i.e. pSTAT3 (Fig. 2l), it suggests that loss of leptin action on glucose is not due to failed leptin action on these neurons.

As requested, we have added feeding and body weight data, shown below, in the revised version as Supplementary Fig. 6.

10) For Figure 3, control vector transduced animals should be included.

Response: As requested, we have provided pictures with control viral delivery in both AgRP and Vgat-Cre mice (shown below) and also in new Fig. 3.

11) For Figure 5d regarding the AgRP-DTR::vgat-Cre mice, my understanding was that they crossed AgRP-Cre with iDTR mice to get heterozygous AgRP-DTR mice, which was subsequently crossed with Vgat-Cre mice. If so, vGat neurons might also express DTR. Please clarify the offsprings used in this study, and provide detailed information regarding how to screen the animals.

Response: The AgRP-DTR mice are a transgenic line (39), which expresses DTR specifically in AgRP neurons without a need of Cre-mediated genomic cleavage. Therefore, in AgRP-DTR::Vgat-Cre mice, DTX will only lesion AgRP neurons but other GABAergic neurons will remain intact.

References

1. A. J. Sipols, D. G. Baskin, M. W. Schwartz, Effect of intracerebroventricular insulin infusion on diabetic hyperphagia and hypothalamic neuropeptide gene expression. *Diabetes* **44**, 147-151 (1995).
2. C. J. Ramnanan, D. S. Edgerton, A. D. Cherrington, Evidence against a physiologic role for acute changes in CNS insulin action in the rapid regulation of hepatic glucose production. *Cell metabolism* **15**, 656-664 (2012).

3. K. W. Williams *et al.*, Segregation of acute leptin and insulin effects in distinct populations of arcuate proopiomelanocortin neurons. *The Journal of neuroscience : the official journal of the Society for Neuroscience* **30**, 2472-2479 (2010).
4. B. C. Borges, X. Han, S. J. Allen, D. Garcia-Galiano, C. F. Elias, Insulin signaling in LepR cells modulates fat and glucose homeostasis independent of leptin. *Am J Physiol Endocrinol Metab* **316**, E121-E134 (2019).
5. J. P. German *et al.*, Leptin activates a novel CNS mechanism for insulin-independent normalization of severe diabetic hyperglycemia. *Endocrinology* **152**, 394-404 (2011).
6. D. Oberlin, C. Buettner, How does leptin restore euglycemia in insulin-deficient diabetes? *J Clin Invest* **127**, 450-453 (2017).
7. K. Hanaki, D. J. Becker, S. A. Arslanian, Leptin before and after insulin therapy in children with new-onset type 1 diabetes. *J Clin Endocrinol Metab* **84**, 1524-1526 (1999).
8. E. H. Hathout *et al.*, Changes in plasma leptin during the treatment of diabetic ketoacidosis. *J Clin Endocrinol Metab* **84**, 4545-4548 (1999).
9. P. J. Havel *et al.*, Marked and rapid decreases of circulating leptin in streptozotocin diabetic rats: reversal by insulin. *Am J Physiol* **274**, R1482-1491 (1998).
10. W. I. Sivitz *et al.*, Plasma leptin in diabetic and insulin-treated diabetic and normal rats. *Metabolism* **47**, 584-591 (1998).
11. M. Y. Wang *et al.*, Leptin therapy in insulin-deficient type I diabetes. *Proceedings of the National Academy of Sciences of the United States of America* **107**, 4813-4819 (2010).
12. X. Yu, B. H. Park, M. Y. Wang, Z. V. Wang, R. H. Unger, Making insulin-deficient type 1 diabetic rodents thrive without insulin. *Proceedings of the National Academy of Sciences of the United States of America* **105**, 14070-14075 (2008).
13. J. P. German *et al.*, Leptin deficiency causes insulin resistance induced by uncontrolled diabetes. *Diabetes* **59**, 1626-1634 (2010).
14. N. Chinooswong, J. L. Wang, Z. Q. Shi, Leptin restores euglycemia and normalizes glucose turnover in insulin-deficient diabetes in the rat. *Diabetes* **48**, 1487-1492 (1999).
15. D. K. Sindelar *et al.*, Low plasma leptin levels contribute to diabetic hyperphagia in rats. *Diabetes* **48**, 1275-1280 (1999).
16. T. Fujikawa *et al.*, Leptin engages a hypothalamic neurocircuitry to permit survival in the absence of insulin. *Cell metabolism* **18**, 431-444 (2013).
17. T. Fujikawa, J. C. Chuang, I. Sakata, G. Ramadori, R. Coppari, Leptin therapy improves insulin-deficient type 1 diabetes by CNS-dependent mechanisms in mice. *Proceedings of the National Academy of Sciences of the United States of America* **107**, 17391-17396 (2010).
18. C. Zhu *et al.*, Profound and redundant functions of arcuate neurons in obesity development. *Nat Metab* **2**, 763-774 (2020).
19. L. Vong *et al.*, Leptin Action on GABAergic Neurons Prevents Obesity and Reduces Inhibitory Tone to POMC Neurons. *Neuron* **71**, 142-154 (2011).
20. Y. Xu, J. T. Chang, M. G. Myers, Jr., Y. Xu, Q. Tong, Euglycemia Restoration by Central Leptin in Type 1 Diabetes Requires STAT3 Signaling but Not Fast-Acting Neurotransmitter Release. *Diabetes*, (2016).
21. Y. Lee, M. Y. Wang, X. Q. Du, M. J. Charron, R. H. Unger, Glucagon receptor knockout prevents insulin-deficient type 1 diabetes in mice. *Diabetes* **60**, 391-397 (2011).
22. N. Damond *et al.*, Blockade of glucagon signaling prevents or reverses diabetes onset only if residual beta-cells persist. *Elife* **5**, (2016).
23. R. J. Perry *et al.*, Leptin reverses diabetes by suppression of the hypothalamic-pituitary-adrenal axis. *Nature medicine* **20**, 759-763 (2014).

24. R. J. Perry, K. F. Petersen, G. I. Shulman, Pleiotropic effects of leptin to reverse insulin resistance and diabetic ketoacidosis. *Diabetologia* **59**, 933-937 (2016).
25. G. J. Morton, T. H. Meek, M. E. Matsen, M. W. Schwartz, Evidence against hypothalamic-pituitary-adrenal axis suppression in the antidiabetic action of leptin. *J Clin Invest* **125**, 4587-4591 (2015).
26. T. H. Meek, G. J. Morton, The role of leptin in diabetes: metabolic effects. *Diabetologia* **59**, 928-932 (2016).
27. Q. Wu, M. P. Boyle, R. D. Palmiter, Loss of GABAergic signaling by AgRP neurons to the parabrachial nucleus leads to starvation. *Cell* **137**, 1225-1234 (2009).
28. F. Meng *et al.*, New inducible genetic method reveals critical roles of GABA in the control of feeding and metabolism. *Proceedings of the National Academy of Sciences of the United States of America* **113**, 3645-3650 (2016).
29. Q. Wu, B. B. Whiddon, R. D. Palmiter, Ablation of neurons expressing agouti-related protein, but not melanin concentrating hormone, in leptin-deficient mice restores metabolic functions and fertility. *Proc Natl Acad Sci U S A* **109**, 3155-3160 (2012).
30. Q. Wu, M. P. Howell, R. D. Palmiter, Ablation of neurons expressing agouti-related protein activates fos and gliosis in postsynaptic target regions. *J Neurosci* **28**, 9218-9226 (2008).
31. R. G. P. Denis *et al.*, Palatability Can Drive Feeding Independent of AgRP Neurons. *Cell Metab* **25**, 975 (2017).
32. S. Luquet, C. T. Phillips, R. D. Palmiter, NPY/AgRP neurons are not essential for feeding responses to glucoprivation. *Peptides* **28**, 214-225 (2007).
33. G. J. Morton, T. H. Meek, M. E. Matsen, M. W. Schwartz, Evidence against hypothalamic-pituitary-adrenal axis suppression in the antidiabetic action of leptin. *J Clin Invest* **2015**, (2015).
34. P. Zouhar *et al.*, UCP1-independent glucose-lowering effect of leptin in type 1 diabetes: only in conditions of hypoleptinemia. *American journal of physiology. Endocrinology and metabolism* **318**, E72-E86 (2020).
35. Y. Xu, Q. Tong, Central leptin action on euglycemia restoration in type 1 diabetes: Restraining responses normally induced by fasting? *The international journal of biochemistry & cell biology*, (2016).
36. G. H. Kim *et al.*, Leptin recruits Creb-regulated transcriptional coactivator 1 to improve hyperglycemia in insulin-deficient diabetes. *Molecular metabolism* **4**, 227-236 (2015).
37. H. C. Denroche *et al.*, Leptin therapy reverses hyperglycemia in mice with streptozotocin-induced diabetes, independent of hepatic leptin signaling. *Diabetes* **60**, 1414-1423 (2011).
38. H. Fenselau *et al.*, A rapidly acting glutamatergic ARCPVH satiety circuit postsynaptically regulated by alpha-MSH. *Nature neuroscience* **20**, 42-51 (2017).
39. S. Luquet, F. A. Perez, T. S. Hnasko, R. D. Palmiter, NPY/AgRP neurons are essential for feeding in adult mice but can be ablated in neonates. *Science* **310**, 683-685 (2005).

Reviewers' Comments:

Reviewer #1:

Remarks to the Author:

The authors have addressed my concerns

Reviewer #2:

Remarks to the Author:

The authors have clearly spent a lot of time considering their responses and broadly speaking, have done so to the points I raised.

Reviewer #3:

Remarks to the Author:

The authors have addressed the comments that I (and others) had raised and their responses to my queries have been made adequately. This has involved some clarifications added to the text, which has improved the overall narrative and readability of the manuscript. Additional experimental data have also been added in response to other reviewers and this has also clarified some key points. I am of the opinion that the work presented is original and convincing with respect to identifying a population of leptin-sensitive ARC GABA neurons that play an important role in leptin-mediated nutrient sensing, which becomes dysfunctional and leads to an aberrant CRR that contributes to chronic hyperglycemia in these rodent models of T1D.

Reviewer #4:

Remarks to the Author:

The authors improved the manuscript while my original questions were not substantially addressed.

For the original questions#1, plasma leptin should be detected in the experimental conditions, which is also helpful to answer the questions raised by the Reviewer 1.

For the original question#6, although previously reported, additional data in experimental conditions in this study should also be provided, which should be easily performed.

For the original question#7, the authors claimed that they only used the data from the ARC only animals, which was pretty difficult. How about the data from the animals with wider infection including the ARC and VMH and other adjacent regions?

The original question#11 was not well addressed, as one caveat for the AgRP-DTR::vgat-cre mice used in this study was that vGat neurons would also express the DTR when AgRP-DTR crossed vGAT-Cre mice, which should be excluded.

Responses to reviewers' comments

We are pleased that the first 3 reviewers have satisfied with our responses and there are some remaining concerns from reviewer 4. The following are our point-to-point responses to these comments.

1) For the original questions#1, plasma leptin should be detected in the experimental conditions, which is also helpful to answer the questions raised by the Reviewer 1.

Response: As requested, to demonstrate leptin changes in T1D, we have measured blood leptin levels in mice shown in Fig. 1a-d. As expected leptin levels were dramatically reduced in T1D compared to controls (Figure on the right). Also as expected, i.c.v. leptin treatment failed to increase leptin levels in blood.

2) For the original question#6, although previously reported, additional data in experimental conditions in this study should also be provided, which should be easily performed.

Response: the original question #6 is “Continued for the Figure1, what about leptin control of food intake and body weight in T1D? which should be performed accompanied with glucose detection.” As we discussed in our previous response, this experiment has to be performed under fasting conditions for glucose and c-Fos responses, so it is not feasible to detect leptin effects on feeding and body weight. The leptin effect on feeding and body weight has been provided in the Supplementary data 6.

3) For the original question#7, the authors claimed that they only used the data from the ARC only animals, which was pretty difficult. How about the data from the animals with wider infection including the ARC and VMH and other adjacent regions?

Response: As requested, we have shown 3 study subjects with missed injections and the injection patterns and glucose levels were shown in the figure on the right. These mice all have miss-targeted VMH neurons.

4) The original question#11 was not well addressed, as one caveat for the AgRP-DTR::vgat-cre mice used in this study was that vGat neurons would also express the DTR when AgRP-DTR crossed vGAT-Cre mice, which should be excluded.

Response: AgRP-DTR is a knockin line (to the AgRP locus) and DTR will not express in vGAT-Cre neurons due to Cre expression.

		Glucose (mM)						
	ID#	0min	15min	30min	60min	90min	2h	4h
Saline	1402	8.7	8	7	5.5	4.9	3.5	2.3
CNO	1402	7.8	8.4	7.9	6.3	4.7	4.7	3.1
Saline	1406	5.4	5.1	4.5	4.3	3.9	3.3	2.1
CNO	1406	4	6.5	8.3	9.8	10	8.4	7
Saline	1409	6.2	5.8	4.7	4.3	3.6	3.4	2.5
CNO	1409	7.4	9.4	10.5	11.6	9.4	8.5	4.3

Reviewers' Comments:

Reviewer #4:

Remarks to the Author:

The authors addressed my concerns.